EMBO
Molecular Medicine

# Evidence that G-quadruplexes form in pathogenic fungi and represent promising antifungal targets

Georgie Middleton[1,2,14], Fuad O Mahamud [3,14], Isabelle S R Storer [1,2,14], Abigail Williams-Gunn[1,2], Finn Wostear[1], Alireza Abdolrasouli[4,5], Elaine Barclay[6], Alice Bradford[1], Oliver Steward [1], Silke Schelenz[7], James McColl[8], Bertrand Lézé[9], Norman van Rhijn[10], Alessandra da Silva Dantas [11], Takanori Furukawa[10,12], Derek Warren[1,13], Zoë A E Waller [3✉] & Stefan Bidula [1,2✉]

## Abstract

**Fungi are estimated to cause the death of almost 4 million people annually, and we urgently need new drug targets to overcome antifungal resistance. We found that four-stranded nucleic acid structures called G-quadruplexes (G4s) could form within the critical priority fungal pathogen *Aspergillus fumigatus*. Sequences with the potential to form G4s could be found in genes involved in fungal growth, virulence, and drug resistance. This included *cyp51A*, which encodes the target of azoles. Notably, we observed the formation of both canonical and unusual acid-stabilised G4s in these sequences. We found that PhenDC3 (a G4-stabilising ligand) could refold DNA into antiparallel G4 structures in *cyp51A* that were associated with decreased transcription. PhenDC3 also had potent fungistatic activity, prevented germination, synergised with the antifungal amphotericin B in vitro and in vivo, and displayed low genotoxicity and cytotoxicity towards human cells. Interestingly, PhenDC3 had greater antifungal activity towards the pan-azole-resistant *A. fumigatus* TR34/L98H isolate, and another G4-stabiliser, pyridostatin, killed multi-drug-resistant *Candida auris*. Taken together, G4s represent a promising target for the development of antifungals with novel mechanisms of action.**

**Keywords** *Aspergillus*; *Candida*; G-quadruplex; Antifungal; DNA
**Subject Categories** Microbiology, Virology & Host Pathogen Interaction; RNA Biology

## Introduction

Pathogenic fungi represent a threat to humanity and contribute to the loss of almost 4 million lives annually (Denning, 2024). This would be an alarming statistic on its own, but fungal infections also devastate critical food crops (e.g., cereals, wheat, rice) and significantly impact biodiversity (Case et al, 2022). With an increase in the utilisation of immunosuppressive medical interventions, increased incidence of pandemic-level infections, and global warming; the frequency of life-threatening fungal infections is likely to rise (Seagle et al, 2021). Worryingly, this rise coincides with an emergence of drug resistance, which has prompted the World Health Organization (WHO) to respond and compile a list of priority fungal pathogens (Fisher and Denning, 2023). This increase in drug resistance has been found to be associated with the overuse of both azole pesticides and more recently, dihydroorotate dehydrogenase (DHODH) inhibitors, in agricultural practices (Bromley et al, 2014; Fraaije et al, 2020; Rhodes et al, 2022; Snelders et al, 2012; van Rhijn et al, 2024). Due to a limited number of effective antifungal agents and the emergence of resistance, novel antifungals with an alternative mechanism of action are essential.

G-quadruplexes (G4s) are non-canonical four-stranded secondary structures that form within guanine-rich regions of DNA and RNA (Varshney et al, 2020). Rather than Watson-Crick base pairing, four guanines can associate with one another via Hoogsteen hydrogen bonding to form G-tetrads. These G-tetrads are connected by loops of nucleotides and can π-π stack to form the G4 structure, with the number of G-tetrads often proportional to G4 stability (Mergny et al, 2005a). Alternatively, another non-canonical structure, the i-motif (iM), can form in the cytosine-rich complementary strand through hemi-protonated cytosine-cytosine base pairing, and they are preferentially stabilised by acidic pH. Notably, the formation of G4s and iMs is interdependent, so when one structure is stabilised, the other is destabilised and G4/iM

[1]School of Chemistry, Pharmacy and Pharmacology, University of East Anglia, Norwich NR4 7TJ, UK. [2]Centre for Microbial Interactions, Norwich Research Park, Norwich NR4 7UG, UK. [3]School of Pharmacy, University College London, London WC1N 1AX, UK. [4]Department of Microbiology, King's College Hospital NHS Foundation Trust, London SE5 9RS, UK. [5]UK Health Security Agency National Mycology Reference Laboratory, Southmead Hospital, Bristol, UK. [6]John Innes Centre Bioimaging Facility, Norwich Research Park, Norwich NR4 7UH, UK. [7]School of Immunology and Microbial Sciences, King's College London, London WC2R 2LS, UK. [8]Henry Wellcome Laboratory for Cell Imaging, University of East Anglia, Norwich NR4 7TJ, UK. [9]Faculty of Science, University of East Anglia, Norwich NR4 7TJ, UK. [10]Manchester Fungal Infection Group, University of Manchester, Manchester M13 9PL, UK. [11]School of Dental Sciences, Newcastle University, Newcastle NE1 7RU, UK. [12]School of Health and Life Sciences, Teesside University, Middlesbrough TS1 3BX, UK. [13]Department of Comparative Biomedical Sciences, The Royal Veterinary College, London, UK. [14]These authors contributed equally: Georgie Middleton, Fuad O Mahamud, Isabelle S R Storer. ✉E-mail: z.waller@ucl.ac.uk; s.bidula@uea.ac.uk

formation dynamics should be considered as interlinked (Brown et al, 2017; King et al, 2020).

Due to the enrichment of G4s/iMs in regulatory regions of the genome, such as the promoters, untranslated regions, and introns, their roles in regulating fundamental biological processes such as transcription, translation, alternative splicing, and epigenetics have become abundantly apparent (Georgakopoulos-Soares et al, 2022; Li et al, 2017; Varshney et al, 2020). G4-ligands, such as PhenDC3 and pyridostatin (PDS), have been designed to specifically stabilise G4s within the genome with negligible interaction to B-DNA (De Cian et al, 2007; Rodriguez et al, 2008). Although much investment has been placed in the therapeutic potential of G4-ligands in the treatment of cancer, there is growing evidence of the antibacterial, antiviral, and antiparasitic possibilities of G4-ligands (Cebrián et al, 2022; Harris et al, 2018; Ruggiero and Richter, 2018). However, the antimicrobial potential of targeting fungal G4s has been poorly explored.

To date, there have been some studies exploring the roles of these non-canonical structures in the model organisms *Saccharomyces cerevisiae* and *Schizosaccharomyces pombe* (Hershman et al, 2008; Lopes et al, 2011; Obi et al, 2020; Paeschke et al, 2011). However, these organisms were mainly used as a proxy to understand human biology and disease. Here, we provide the first comprehensive study highlighting the formation of G4s in a pathogenic fungus and instead describe the therapeutic potential of targeting non-canonical nucleic acid structures in both drug-susceptible and drug-resistant critical priority fungal pathogens.

## Results

### The G4-ligands PhenDC3 and PDS display potent antifungal activity against *Aspergillus* spp. and *Candida* spp

Given the increased prevalence of antifungal resistance, identifying drugs with novel mechanisms of action is of utmost importance. To explore the antifungal potential of G4-ligands, we tested the antifungal activity of PhenDC3 and PDS, two commonly used G4-ligands, against a panel of *Aspergillus* and *Candida* species.

PhenDC3 potently inhibited the metabolic activity of a range of drug-susceptible, azole-resistant, and cryptic *A. fumigatus* isolates. The $MIC_{90}$ for the azole-susceptible *A. fumigatus* clinical isolate 22M7007854 was 1.56 μM (Fig. 1A). Surprisingly, PhenDC3 had increased activity against 22M7004177 ($MIC_{90} = 0.78$ μM) and $TR_{34}/L98H$ ($MIC_{90} = 0.78$ μM); itraconazole and pan azole-resistant isolates, respectively. PhenDC3 also retained its activity against two common *A. fumigatus* laboratory strains, CEA10 ($MIC_{90} = 3.13$ μM) and A1160+ ($MIC_{90} = 3.13$ μM). Conversely, PDS had activity against all the isolates tested, but its efficacy was lower compared to PhenDC3 ($MIC_{90} = 12.50$ μM for 22M7007854, 22M7004177, and $TR_{34}/L98H$; $MIC_{90} = 6.25$ μM for CEA10 and A1160 + ; Fig. 1B).

Next, we investigated the effects of PhenDC3 against non-Fumigati and cryptic Fumigati species to see if our observations were transferable to related fungi. PhenDC3 appeared to lose activity against the non-Fumigati species *A. flavus*, *A. brasiliensis*, and *A. niger*, but it was still inhibitory against the cryptic Fumigati species *A. udagawae* ($MIC_{90} = 6.25$ μM) and *A. hiratsukae*

($MIC_{90} = > 12.5$ μM), which are genetically very similar to *A. fumigatus* (Fig. 1C). PDS was also inactive against *A. flavus* but displayed activity against *A. niger* and *A. brasiliensis* ($MIC_{90} = 12.5$ μM), *A. hiratsukae* ($MIC_{90} = 6.25$ μM), and *A. udagawae* ($MIC_{90} = 12.5$ μM) (Fig. 1D).

Finally, we assessed the antifungal activity of both PhenDC3 and PDS against *Candida* spp. commonly associated with human infections; *C. albicans*, *C. glabrata* (*Nakaseomyces glabratus*), and *C. auris* (*Candidozyma auris*), to determine if G4-ligands had activity against fungi more distantly related to Aspergilli. Surprisingly, PhenDC3 displayed poor activity against *C. auris* ($MIC_{50} = 30.03$ μM) and no activity against the other *Candida* spp. tested (Fig. 1E). However, PDS was found to be more active against these yeasts compared to filamentous fungi and displayed potent activity ($MIC_{50} = 5.48$ μM, 3.94 μM, and 5.94 μM for *C. auris*, *C. glabrata*, and *C. albicans*, respectively; Fig. 1F). A full list of potencies can be found in Appendix Table S1.

### PhenDC3 and PDS are fungistatic and fungicidal, respectively

To determine if PhenDC3 and PDS were fungicidal or fungistatic, we performed a killing assay using CEA10. PhenDC3 was found to be fungistatic, whilst PDS was fungicidal, with MIC values of 1.56 μM and 12.5 μM being recorded for PhenDC3 and PDS, respectively (Fig. 2A). Notably, the phenotype of the colonies recovered 48 h after PhenDC3 treatment was smaller and less sporulated than untreated colonies. Colonies with a similar phenotype were also recovered following treatment with the known fungicidal antifungal, amphotericin B (Fig. 2B). In addition, to explore how quickly G4-ligands could exert their antifungal effects, we performed a kill-curve analysis. We found that G4-ligands had a fast mechanism of action: MIC concentrations of PhenDC3 and PDS prevented >90% of growth after 24 h of outgrowth following 8 and 1 h of treatment, respectively (Fig. 2C,D).

### PhenDC3 can completely prevent hyphal formation and causes spores to swell excessively

The rapid mechanism of action of PhenDC3 and PDS was mirrored in their ability to prevent *A. fumigatus* from germinating, and both compounds could inhibit germination in a concentration-dependent manner (Fig. 3A,B). PhenDC3 also had activity against growing germlings but no activity against mature biofilms at the concentrations tested (Appendix Fig. S1). Non-germinated conidia treated with PDS did not swell, which may be attributed to the fast killing exhibited by PDS (Fig. 3C). Rather than germinating, conidia treated with PhenDC3 became excessively swollen: the mean untreated spore area was 2.66 μm ± 0.16 SEM, whilst the mean spore area when treated with 0.39 μM PhenDC3 was 34.38 μm ± 2.30 SEM (Fig. 3D). However, a small subset of conidia did not appear to swell or germinate. Images of conidia treated for 24 h with PDS and PhenDC3 are found in Fig. 3E. Scanning electron microscopy confirmed that PhenDC3 spores appeared to be excessively swollen (Fig. 3F). Moreover, PhenDC3 treatment caused the spore surface to appear far more 'cracked' compared to control, whilst PDS treatment caused the spore surface to appear more 'wrinkled'.

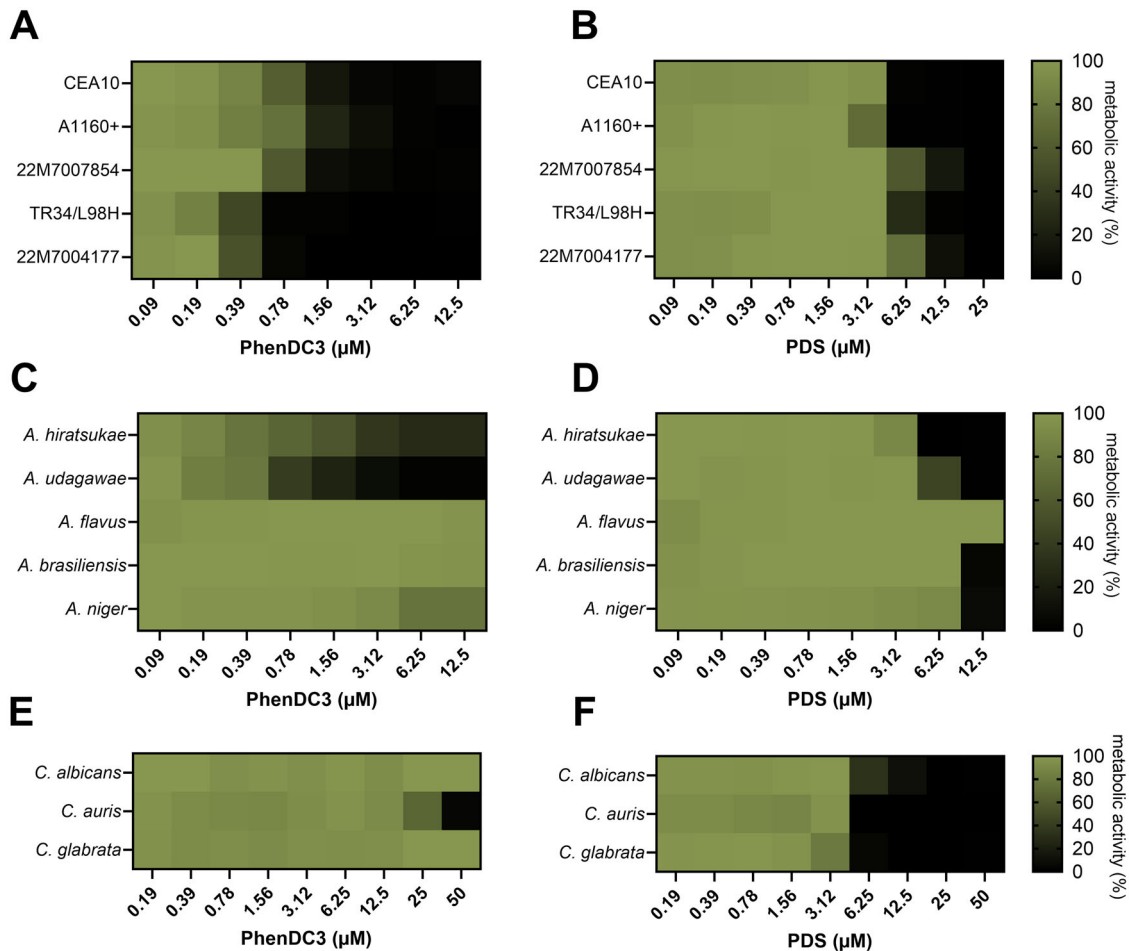

**Figure 1. PhenDC3 and PDS inhibit the metabolism of drug-susceptible and drug-resistant fungal pathogens.**

Drug susceptible (22M7007854) and itraconazole-resistant (22M7004177) *A. fumigatus* clinical isolates, laboratory strain A1160 +, 'wildtype' strain CEA10, and the pan azole-resistant *A. fumigatus* isolate (TR$_{34}$/L98H) were treated with a concentration range of (**A**) PhenDC3 or (**B**) PDS for 48 h at 37 °C. Non-*fumigatus Aspergillus* strains treated with (**C**) PhenDC3 or (**D**) PDS for 48 h. *Candida* species were treated with the indicated concentrations of (**E**) PhenDC3 and (**F**) PDS for 24 h. Metabolic activity was calculated as a percentage of the untreated control. Experiments are in biological triplicate with each data point consisting of the mean of three technical replicates. Source data are available online for this figure.

## Targeting fungal G4s may have therapeutic potential

The above data demonstrated that PhenDC3 possessed impressive antifungal activity. Next, we explored whether PhenDC3 had the potential to be utilised therapeutically and assessed the synergy of PhenDC3 with current antifungals, the in vivo efficacy of PhenDC3, and the effect of G4-ligands on disease-relevant human cells.

Checkerboard assays performed with CEA10 conidia indicated that PhenDC3 could synergise with amphotericin B (FICI = 0.375), with a combination of 0.39 μM PhenDC3 and 0.125 μg/mL amphotericin B able to fully inhibit germination (Fig. 4A). Meaning that 8-fold lower concentrations of amphotericin B and 4-fold lower concentrations of PhenDC3 could be used in combination. PhenDC3 did not synergise with itraconazole or voriconazole. *Galleria mellonella* larvae (Greater wax-moth) have an innate immune system remarkably like humans and represent a useful model to study fungal infections in vivo (Giammarino et al, 2024).

Uninfected *G. mellonella* larvae which were not injected, injected with PBS alone, or injected with 2.5 mg/kg PhenDC3 (equivalent to 4.91 μM) or 2.5 mg/kg amphotericin B (equivalent to 4.51 μM) displayed ≥90% viability over the 7-day experimentation period (Fig. 4B,C). Larvae inoculated with $1 \times 10^6$ *A. fumigatus* conidia alone displayed 89.6% mortality after 7 days. In larvae infected with *A. fumigatus*, 2.5 mg/kg amphotericin B, delayed larval death (50% versus 34% survival at day 4 for amphotericin B and control, respectively) and caused a small decrease in overall mortality at day 7 (9.6%; $P > 0.05$). Notably, 0.25 mg/kg PhenDC3 (equivalent to 491 nM) also delayed larval death (68% versus 34% survival at day 4 for PhenDC3 and control, respectively) but reduced mortality at day 7 by 52% ($P = 0.0018$). This concentration was more effective than 2.5 mg/kg PhenDC3 (36% survival at day 7; $P = 0.0254$).

As we found that PhenDC3 and amphotericin B could synergise in vitro, we tested whether this combination retained efficacy in vivo. For these experiments, we combined 10-fold less PhenDC3 with 100-fold less amphotericin B (both at a concentration of

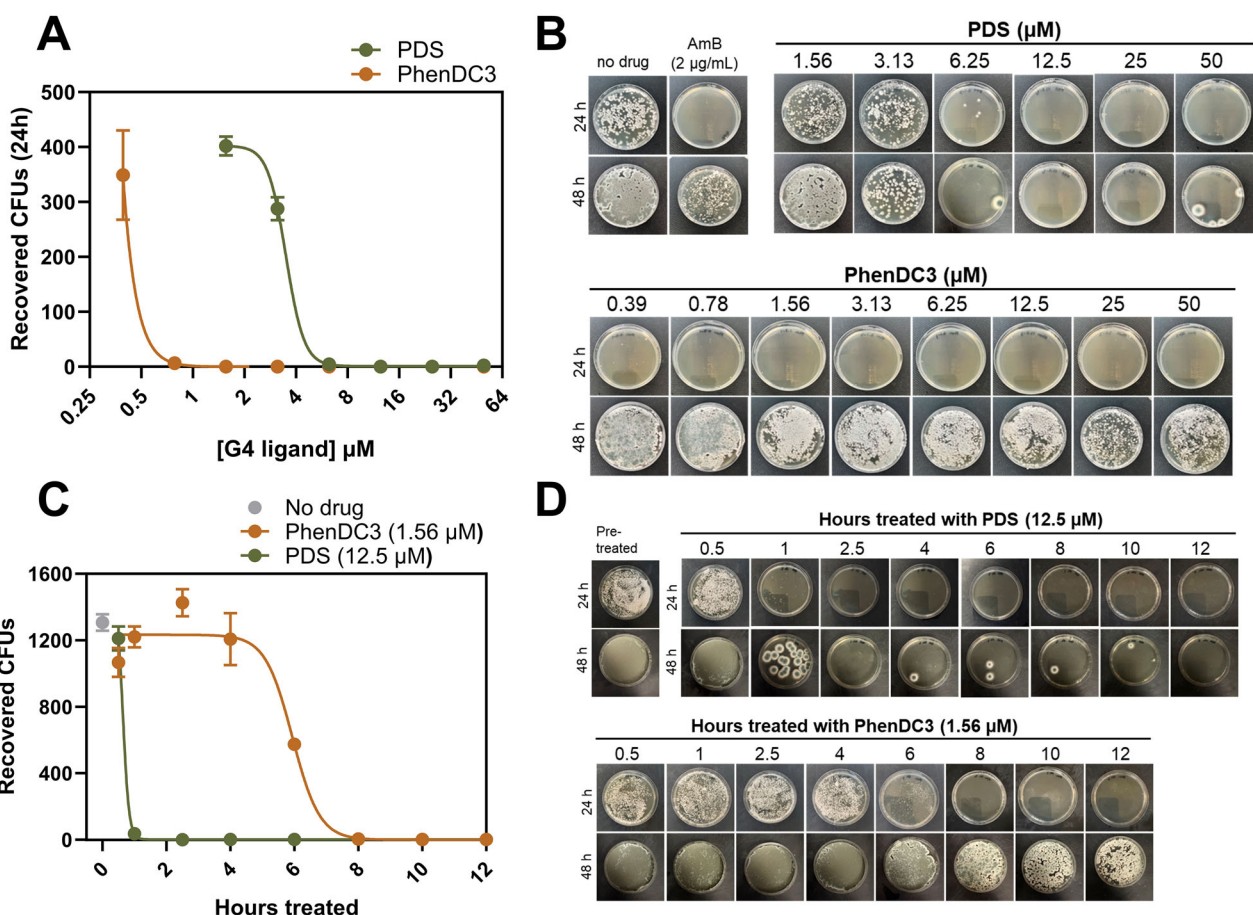

**Figure 2. PhenDC3 and PDS exhibit fungistatic and fungicidal activity, respectively, within 8 h of treatment.**

(A) Viability counts and (B) colony phenotypes of *A. fumigatus* conidia (CEA10) after 24 h of exposure to the indicated range of PDS and PhenDC3. SDA plates were incubated for 24 h and 48 h h post exposure. Counts are the mean of three biological replicates. (C) Viability counts and (D) colony phenotypes of *A. fumigatus* conidia (CEA10) after the indicated time of exposure MIC$_{90}$ concentrations of PDS and PhenDC3. SDA plates were incubated for 24 h and 48 h h post exposure. Counts are the mean of three biological replicates. Error bars represent the SEM. Source data are available online for this figure.

0.025 mg/kg; equivalent to 49.1 nM and 45.1 nM, respectively). We found that this combination was equally as efficacious as amphotericin B alone at 2.5 mg/kg in this study (20% survival after 7 days), but was more effective at delaying larval death (68% versus 50% survival at day 4 for PhenDC3 and amphotericin B combined, versus amphotericin B alone).

PhenDC3 was highly effective against *A. fumigatus*, but G4s can be found in all organisms, including humans, so we next assessed the cytotoxic effects of G4-ligands against human cells. We assessed the cytotoxicity of PhenDC3 and PDS against the A549 type II lung alveolar epithelial cell line (representative of cells in the lung) and primary human vascular smooth muscle cells (representative of the predominant cell type in arteries and veins). There was no cytotoxicity observed in A549 cells for PhenDC3 up to 50 μM and the ED$_{50}$ for PDS was ~50 μM (Fig. 5A; Appendix Fig. S2). Conversely, PhenDC3 had much lower toxicity against smooth muscle cells compared to PDS (ED$_{50}$ of 72.8 μM for PhenDC3 versus 13.1 μM for PDS) (Fig. 5B). Stabilising G4s has the potential to cause DNA damage and genome instability (De Magis et al, 2019). We assessed the amount of DNA damage caused by PhenDC3 in the primary vascular smooth muscle

cells by quantifying the number of γH2AX foci, indicative of double-strand breaks (Kuo and Yang, 2008). We found that there was no significant difference in the number of γH2AX foci between control cells (mean 0.75 foci ± 0.42 SEM per nucleus) and cells treated with 25 μM (mean 0.17 foci ± 0.06 SEM per nucleus) or 50 μM (mean 0.40 foci ± 0.19 SEM per nucleus) PhenDC3 (Fig. 5C,D). PDS disrupted cell structure and caused cells to rupture, and DNA damage could not be quantified accurately in these cells (Appendix Fig. S3).

## G4s can form in A. fumigatus RNA and are promoted by PhenDC3

The frequency of G4s in the *A. fumigatus* reference genome, Af293, was found to be comparable to the G4-frequency observed in humans at around 1.5 per 1000 base pairs when using the default G4Hunter settings (Appendix Table S2). This is far higher than our initial estimations as these parameters account for the detection of non-canonical G4s which were filtered out in our previous analyses (Warner et al, 2021). From our experiments, it was clear that stabilising G4s had a profound biological effect, but the formation

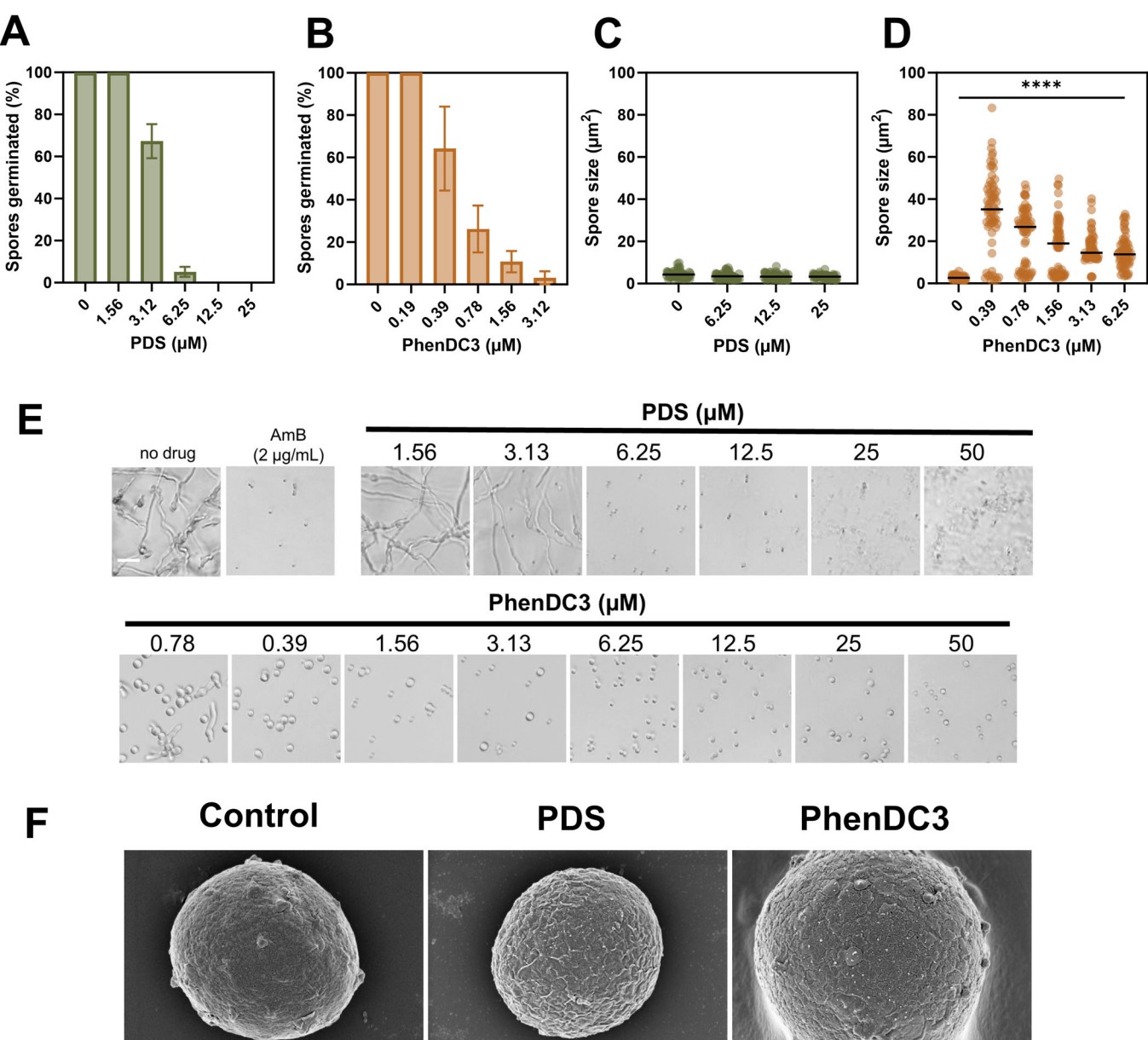

**Figure 3. PhenDC3 has a profound inhibitory effect on *A. fumigatus* germination and causes spores to swell excessively.**

The percentage of conidia which germinated after 24 h of exposure at 37 °C to (A) PDS or (B) PhenDC3. The mean ± SEM of three biological replicates is shown. The area of conidia which did not germinate was measured using Fiji 1.53t ($n = 60$) after 24 h exposure to (C) PDS or (D) PhenDC3. Statistical significance was determined by a Kruskal–Wallis test with Dunn's multiple comparisons correction (H (5) = 174.5, $P < 0.0001$). untreated vs. each PhenDC3 treatment concentration were significantly different ($P < 0.0001$, (****)). (E) Images of *A. fumigatus* conidia treated for 24 h with the indicated concentration of G4-ligand. Scale bar: 20 μm. (F) *A. fumigatus* conidia were treated with 12.5 μM PDS or PhenDC3 for 24 h prior to scanning electron microscopy. The sample was sputter-coated with platinum for 1 min at 10 mA before imaging on a Zeiss Gemini 300 scanning electron microscope and viewed at 2 kV. Scale bars represent 400 nm. Source data are available online for this figure.

of G4s or iMs in *A. fumigatus* RNA and DNA had never been confirmed experimentally. Therefore, we aimed to provide definitive evidence that these non-canonical nucleic acid structures could form in a pathogenic fungus and G4 ligands could influence G4 formation.

To determine RNA G4 formation in live fungi, we used QUMA-1, a dynamic dye that fluoresces when it binds to RNA harbouring G4 structures (Chen et al, 2018). In these experiments, Hoechst 33342 was intended to be used to label the fungal nuclei. However, it did not enter live fungi when incubated for a short period of time, and this was used instead to help outline the fungus and enhance visualisation. QUMA-1 staining was diffuse in untreated *A. fumigatus*, with few foci observed (mean 0.35 ± 0.11 SEM foci per 10 μm²) (Fig. 6A). Notably, PhenDC3 induced an obvious increase in the number of punctate QUMA-1 foci at both 1.56 μM (mean 2.65 ± 0.26 SEM foci per 10 μm²) and 12.5 μM (mean 3.09 ± 0.39

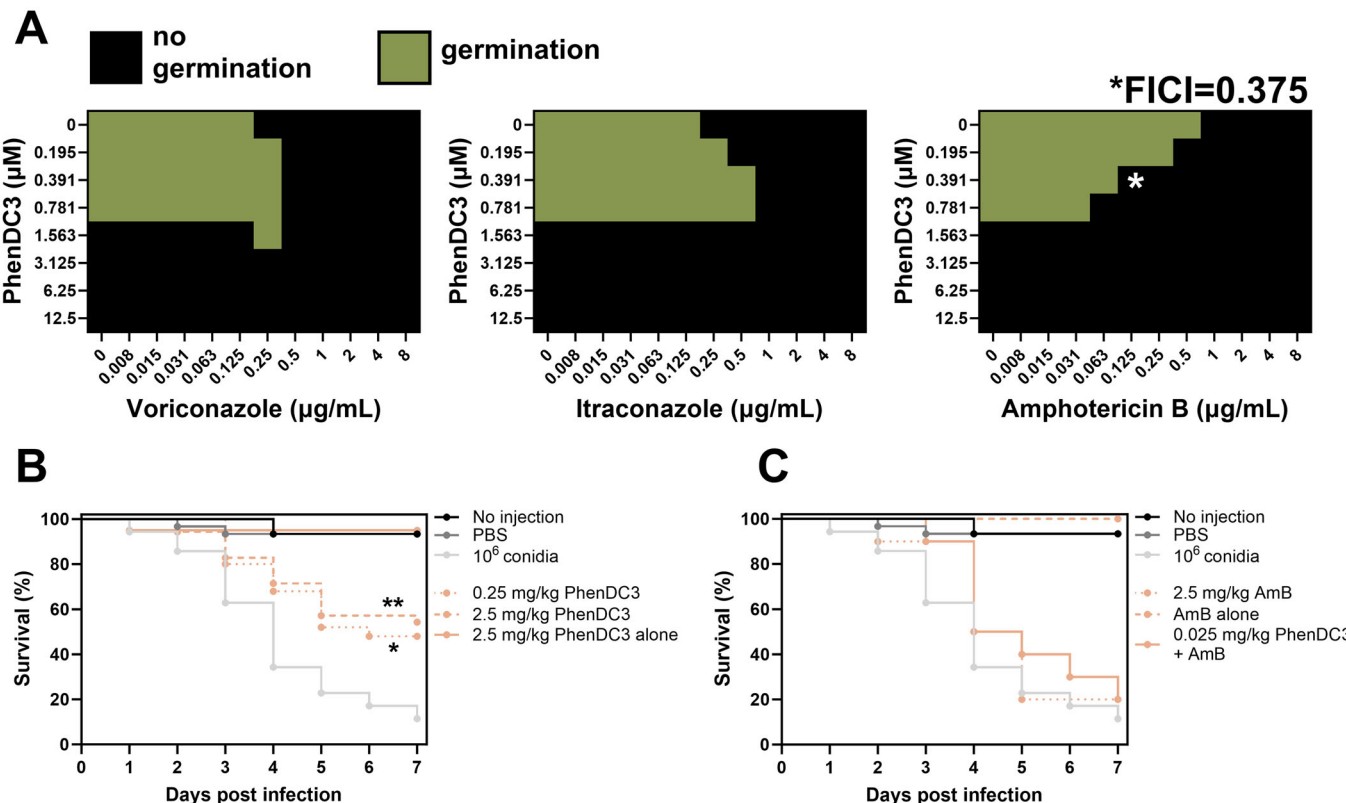

**Figure 4. PhenDC3 synergises with amphotericin B and is protective in a *G. mellonella* infection model.**

(A) *A. fumigatus* conidia were treated with PhenDC3 in combination with the antifungals amphotericin B, itraconazole, or voriconazole at the concentrations indicated for 24 h in a checkerboard assay. The black cells indicate where germination was completely prevented. A fractional inhibitory concentration index (FICI) below 0.5 indicates synergy. (B, C) The percentage survival of *G. mellonella* larvae which were injected with PBS, 2.5 mg/kg amphotericin B (AmB) alone, 2.5 mg/kg PhenDC3 alone, or $1 \times 10^6$ *A. fumigatus* conidia alone was monitored for 7 days. In some groups, larvae infected with $1 \times 10^6$ *A. fumigatus* conidia were treated with 0.25 mg/kg PhenDC3, 2.5 mg/kg PhenDC3, 2.5 mg/kg AmB, or a combination of PhenDC3 and AmB (both at 0.025 mg/kg), and survival was monitored for 7 days. Data are representative of two or three biological replicates with 10 larvae per treatment group (20-30 total). Data for both figures are from the same experiments, but the figure has been split into two for clarity. Statistical significance was determined by Kaplan–Meier survival analyses ($P = 0.0018$ (**) and $P = 0.0254$ (*) for 0.25 mg/kg and 2.5 mg/kg PhenDC3 versus conidial infection alone). Source data are available online for this figure.

SEM foci per 10 μm²) (Fig. 6B), suggesting that PhenDC3 was influencing G4-formation directly in fungi as anticipated. Fungi treated with 1.56 μM PDS exhibited a phenotype comparable to the control, but 12.5 μM PDS prevented the swelling of conidia. It was impossible to determine the foci in the 12.5 μM PDS-treated spores because the spores were dead, and the membrane was more permeable to both dyes.

## G4s have the potential to form in A. fumigatus genes involved in diverse biological processes, ranging from cell wall development to drug resistance

To garner additional insights into where G4s might be forming, sequences with the potential to form G4s were identified in a variety of genes involved in diverse biological processes within *A. fumigatus*. These sequences were annealed in a buffer that promotes G4 formation prior to the addition of thioflavin T, which has previously been shown to increase in fluorescence upon binding to G4 structures (Zhang et al, 2018). As expected, thioflavin T fluorescence increased in the presence of the G4-forming telomeric repeat sequence [GGGTTA]₄ but not in the presence of

the 'B-DNA' sequence which is incapable of forming a G4 (Table 1; Appendix Fig. S4). Exploring the G4-forming sequences found in fungal genes, only those taken from *skn7* and *mp1* were incapable of forming G4s, genes encoding proteins involved in the oxidative stress response and a cell wall mannoprotein, respectively. Notably, G4s were found to form in a range of purified single-stranded DNA oligonucleotides representative of sequences found in genes involved in cell wall formation, dynamics, and development; including those involved in formation of the rodlet layer (*rodA*), melanin (*arp2*), ergosterol synthesis (*cyp51A*, *cyp51B*, *hmg1*, *atrR*, *erg1*, *erg5*), α1,3-glucans and β1,3-glucans (*ags1*, *fks1*) and chitin synthesis (*chsG*).

## The DNA sequences from cyp51A and cyp51B show structural flexibility and can form iMs, G4s and hairpins/duplexes

To investigate G4-formation in more detail, we selected two candidate targets from the thioflavin T analyses. These two targets, *cyp51A* and *cyp51B*, encode two sterol 14-demethylases critically involved in the synthesis of ergosterol in the fungal cell membrane

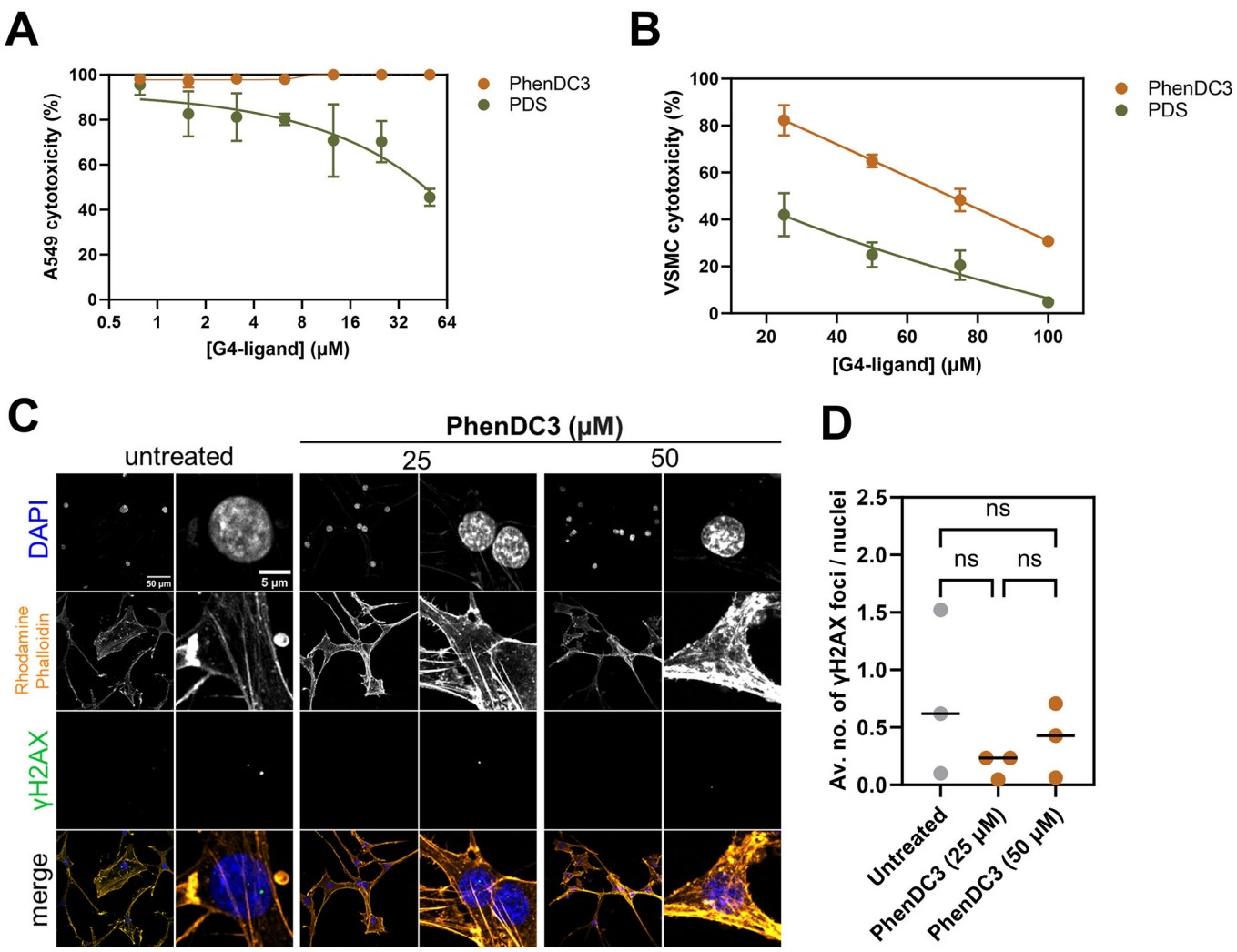

**Figure 5. PhenDC3 is well tolerated by primary smooth vascular cells with negligible DNA damage.**

The viability of (**A**) the human A549 lung adenocarcinoma cell line or (**B**) primary human vascular smooth muscle cells was quantified following treatment with a concentration range of PhenDC3 or PDS for 24 h. Metabolism was measured using resazurin and normalised to the untreated control. The mean ± SEM of three biological replicates is shown. (**C**) Representative confocal microscopy images of primary human vascular smooth muscle cells treated with 25 µM or 50 µM PhenDC3 for 24 h to quantify γH2AX foci, which are indicative of DNA damage. (**D**) The average number of γH2AX foci per nucleus following treatment with 25 µM or 50 µM for 24 h. Images were obtained using a Zeiss LSM 980 laser scanning confocal microscope equipped with an Airyscan 2 detector using the ×20 objective. γH2AX foci were quantified using Fiji 1.53t. Experiments were performed in biological triplicate. Each data point represents the mean of at least three technical replicates. Statistical significance was determined by a Brown-Forsythe and Welch ANOVA test with Dunnett's T3 multiple comparisons test (ns not significant). Source data are available online for this figure.

and represent the target of azole antifungals. Three G4-forming sequences could be found in these genes. These sequences were found in the *cyp51A* coding and 3′-untranslated regions (termed A1-G and A2-G, respectively), and the coding region of *cyp51B* (termed B-G). Here, we biophysically characterised these three sequences and their reverse complement (A1-C, A2-C, and B-C). We found that these DNA sequences displayed a large amount of structural flexibility depending upon the cations present or the pH and were capable of folding into G4s, iMs, and hairpins/duplexes as determined by circular dichroism (CD), thermal difference spectra (TDS) and UV-melting experiments (Table 2).

To identify the full range of structures that these sequences could fold into, we assessed DNA folding in a range of different pH levels. At pH 7.0, A1-G and B-G gave CD spectra with a broad

positive peak at 260 nm and a negative peak at 245 nm (Fig. 7, bottom panel). This type of spectrum could indicate either a parallel G4 or duplex structure. Complementary TDS confirmed hairpin/duplex DNA conformation, rather than G4, with a notable absence of a negative band at 295 nm (Mergny et al, 2005b). UV-melting experiments showed transitions only at 260 nm, in line with hairpin/duplex structures, and not at 295 nm, which would be expected for G4s. Unexpectedly, as the pH of the samples was lowered, the CD spectra of both A1-G and B-G transitioned into antiparallel G4 structures, as indicated by an increased signal at 295 nm in the CD spectra. This change was consistent with a pH-dependent conformational shift, where the degree of protonation of cytosine bases affects the stability and folding of G4 structures (Benabou et al, 2019).

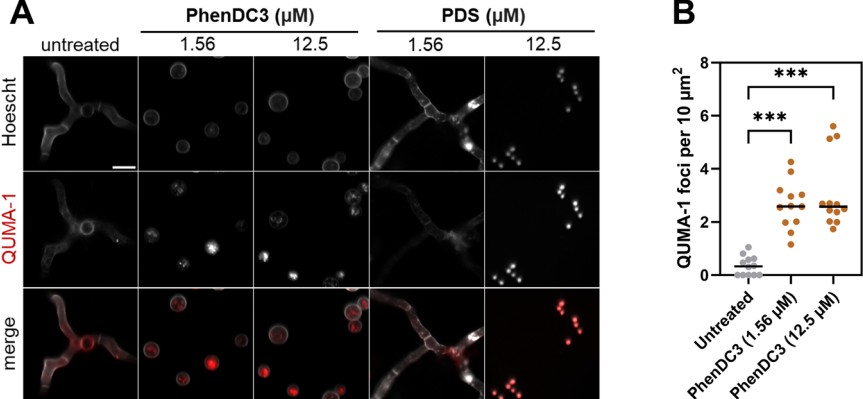

**Figure 6.  G4s form within *A. fumigatus* RNA in vivo, and their formation is promoted by PhenDC3.**

(A) *A. fumigatus* conidia were treated with the indicated concentration of G4-ligand for 18 h at 37 °C. Treated cells were washed with PBS and incubated with 1 µM Hoechst 33342 and 1 µM QUMA-1 at 37 °C for 1 h to observe RNA G4. Representative images were captured via confocal microscopy using a ×100 oil objective. Scale bar: 10 µm. (B) The number of QUMA-1 foci per cell per image ($n = 12$ technical replicates from three biological replicates) was quantified using Fiji 1.53t and normalised per area to allow for differences in germination. Statistical significance was determined by a Kruskal–Wallis test with Dunn's multiple comparisons correction (H (2) = 3.47, $P < 0.001$). Untreated vs. PhenDC3 (1.56 µM) or (12.5 µM) were significantly different ($P < 0.001$, (***)). Source data are available online for this figure.

These hairpin/duplex-to-G4 transitions were supported by the UV-melting experiments (i.e., increased melting temperature of G4 versus duplex in KCl at pH 5.5) and TDS (positive peaks at 240, 255, and 270 nm, and a negative peak at 295 nm). All supporting data for these and the following biophysical experiments can be found in Appendix Figs. S5–S10.

The C-rich sequences A1-C and B-C transitioned between hairpin and iM structures depending on the cationic and pH conditions (Fig. 7, top panel). CD spectra at pH 7.0 showed positive ellipticity at 272 nm, indicative of hairpin/duplex formation. As the pH decreased, the spectra shifted to a maximum at 288 nm, and negative peak at 240 nm, consistent with the formation of an iM structure. The TDS profile for these sequences showed positive peaks at 240 nm and 265 nm with a negative peak at 295 nm, characteristic of iM conformation at pH 5.5. A1-C also exhibited distinct UV melting and annealing profiles at 260 nm at pH 7.0 under different cationic conditions yet only displayed a melting profile at 295 nm in KCl at pH 5.5.

A2-G adopted both hairpin/duplex and G4 conformations under physiological-like conditions (Fig. 7, middle panel). This was characterised by a positive peak at 263 nm, with a shoulder at 280 nm, and a negative peak around 240 nm, indicating the co-existence of two structural forms. This result was further supported by UV-melting data measured at 260 nm and 295 nm. In KCl at pH 7.0, the melting temperature from the data recorded at 260 nm was $40.6 \pm 0.1$ °C, corresponding to the hairpin structure, whereas the melting temperature at 295 nm was significantly higher ($54.2 \pm 0.6$ °C), indicative of the presence of a more thermodynamically stable G4 structure. Sequence, A2-C formed iM structures and was significantly more stable at pH 5.5 compared to pH 7.0.

**PhenDC3 can induce a structural shift from a parallel to an antiparallel G4 in the cyp51A coding region, but PDS cannot**

We examined the interactions of PhenDC3 and PDS on the G-rich sequences A1-G, A2-G, and B-G (Fig. 8A,B). PhenDC3 was found

to cause the hairpin/duplex structures in A1-G and parallel G4 structures in A2-G to transition to antiparallel G4 conformations. The transition from a hairpin/duplex structure to an antiparallel G4 in A1-G was particularly striking. The binding of PhenDC3 was found to be cooperative, with a K value of $32.7 \pm 3.3$ µM and a Hill coefficient of $2.1 \pm 0.9$ when interacting with PhenDC3 (Appendix Fig. S10). Similarly, A2-G showed a comparable binding affinity, with a K value of $32.5 \pm 1.8$ µM and a Hill coefficient of $2.8 \pm 1.5$ for PhenDC3 (no difference between PhenDC3 binding to A1-G or A2-G; $P = 0.9486$). Contrarily, PhenDC3 had a limited effect on the hairpin/duplex structure formed in B-G. Importantly, PDS did not induce any structural alterations in any of the sequences tested. In support of PhenDC3's influence on these *cyp51A* sequences (A1-G and A2-G), PhenDC3 reduced the expression of *cyp51A* by 44% relative to control ($P < 0.05$; Fig. 8C). We were unable to obtain high-quality RNA from the PDS-treated fungi, presumably due to the fast killing by PDS.

## Discussion

This work provided a comprehensive comparison between the impact of the G4-ligands PhenDC3 and PDS on fungal biology. We found that these ligands were both antifungal, but in very different ways. Further, we showed that G4-ligands had the potential to be used therapeutically, but additional in-depth studies are essential to fully realise the potential of G4s as a drug target in fungi. Additionally, we demonstrated the structural flexibility of fungal DNA to form a wide range of secondary structures that are often overlooked when considering the effects of xenobiotics on the biological responses of fungi. This included two unusual G4s in *cyp51A*, which were stabilised by acidic pH. Taken together, we highlight both the potential of G4s as novel drug targets and allude to the biological importance of non-canonical nucleic acid structures within fungal biology.

**Table 1. G4s can form in sequences predicted to form G4s found in genes linked with drug resistance, virulence, and development.**

| Sequence ID | Oligonucleotide sequence | Fluorescence ± SEM (a.u.) | Genomic location |
|---|---|---|---|
| *rodA* | GGGGTAGGGAACAGGGTCCCGGG | 50,100 ± 17 | 3′UTR |
| *arp2* | GGGGAGGGGACCCTTGCATGGAGGGGGGG | 82,140 ± 54 | CDS |
| *ags1* | GGGCTTGAGGGGAACTGGGAGGAGCGGG | 67,230 ± 34 | CDS |
| *cyp51A* | GGGTGGGGGTGATAAACGGGGCATCAGGG | 62,145 ± 13 | 3′UTR |
| *cyp51B* | GGGGTAGGGGGGGCCCAGGG | 34,409 ± 10 | CDS |
| *hmg1* | GGGCGGCAGGGACAGGGGCAACTTCGGG | 148,303 ± 33 | CDS |
| *atrR* | GGGGGAGGGTGCGGGATGCTCTTCGGG | 37,691 ± 7 | CDS |
| *erg1* | GGGCGACGGGGAAGAGGTAGGGAGGG | 71,126 ± 17 | CDS |
| *erg5* | GGGATTAGGGTATGCCTCAGGGTCATGGG | 149,474 ± 14 | CDS |
| *mepB* | GGGCGAGGGCAGTGGGTTCGTGGG | 37,952 ± 5 | CDS |
| *dppV* | GGGACGGCGTTGGGGGGGGGAAGGG | 47,933 ± 12 | Intron |
| *mirB* | GGGTAGCTGAGCTGGGTGGGGAGGG | 36,245 ± 7 | CDS |
| *zrfC* | GGGTTGGGCTGGTCGACATGTGGGCCCGGG | 81,452 ± 19 | CDS |
| *zafA* | GGGCAGGGCAGAGCAGGGCAGGGG | 26,023 ± 6 | Intron |
| *lysF* | GGGACGCGGGGGGCGGCATCGGGAGCGGG | 86,442 ± 9 | CDS |
| *hscA* | GGGGCTGAATTGGGGGGGAAAACGAGACGGG | 56,161 ± 21 | Promoter/5′UTR |
| *aspf2* | GGGGACGGGGGAGGTAGGGAGGG | 87,449 ± 40 | CDS |
| *aspf28* | GGGTTGCGAGGGAAGGAGGGGCGGG | 71,548 ± 13 | Intronic |
| *gstA* | GGGCTGGGTGGCGGCCGCTGGTTGGGCGGG | 56,709 ± 4 | CDS |
| *yap1* | GGGATGGGCTTCCCAAGTCGGGGAGAGGG | 69,030 ± 18 | CDS |
| *skn7* | GGGGGGGGGGGTGGG | 7173 ± 6 | 3′UTR |
| *pes1* | GGGAGCAAGGGGAAATCGCTGAGGGTGGG | 129,161 ± 40 | CDS |
| *atrF* | GGGTATTTGTTTGGGGGGGATGGG | 64,599 ± 9 | CDS |
| *abcB* | GGGGGTGGGCGTTGGGGCGGG | 28,763 ± 5 | 5′UTR |
| *fmpD* | GGGGGGACCCGTCATGGGCGGG | 28,495 ± 8 | CDS |
| *mfsB* | GGGAGACAAGGGGGAGGGGGG | 48,489 ± 6 | 5′UTR |
| *mnt1* | GGGAGAGGGCACAGGGGGG | 49,525 ± 5 | Intronic |
| *hsp70* | GGGACACCACGGGGGGGCGGG | 28,707 ± 14 | CDS |
| *pacC* | GGGGGGGGGGGAATGCTTGGG | 32,214 ± 11 | 5′UTR |
| *ste7* | GGGAGGGACACAAAGAGGGGATAAGGAGGG | 126,709 ± 54 | 5′UTR |
| *bck1* | GGGCCTGGAGGGGGGCCCGGGG | 73,174 ± 7 | CDS |
| *pkaC1* | GGGTTGTGGTGGGTGGAGGGATGGGG | 62,347 ± 4 | CDS |
| *laeA* | GGGATGTGGGACAGGGATTTGGG | 50,899 ± 9 | CDS |
| *gliN* | GGGGGGGCGCTTGGCTGGGAAGGCGGTTGGG | 134,267 ± 16 | CDS |
| *ftmC* | GGGTCTGGAGCGCGGGGTTCGGGGGATCGGG | 66,613 ± 9 | CDS |
| *fks1* | GGAAGGGGCTCGGGCATGGG | 72,302 ± 9 | CDS |
| *mp1* | GGCTCCGGCTCCGGCTCCGG | 9852 ± 22 | CDS |
| *chsG* | GGAGAGCGGGAGGTGGTGAGGG | 38,112 ± 15 | 5′UTR |
| *ftmE* | GGGAGTGGGGATGGCGACGGGGAGCAGGG | 105,384 ± 10 | CDS |
| B DNA | GGCGCTATTCTCTGGTGCTGCT | 8808 ± 6 | N/A |
| Telomere | TTAGGGTTAGGGTTAGGGTTAGGGTT | 100,859 ± 19 | N/A |
| Background | No DNA | 1154 ± 33 | N/A |

Gene sequences with G4-forming potential were identified in genes associated with drug resistance, virulence, and fundamental biological processes in *A. fumigatus*. These sequences (4 μM) were annealed in buffer containing 10 mM sodium cacodylate and 100 mM KCl and incubated with thioflavin T (1 μM), which fluoresces following binding to G4s. The background is representative of thioflavin T fluorescence in the absence of DNA. B-DNA unable to form a G4 was used as a negative control, and the [GGGTTA]₄ telomeric repeat sequence was used as a positive control. Guanines are highlighted to facilitate identification of the G4-forming motifs.

**Table 2. Biophysical characterisation data for the Cyp51 sequences.**

| | Sequences | pH | Cations | UV Spectroscopy melting, annealing temperature (°C) | | | | pH_T | CD Structure | TDS Structure |
|---|---|---|---|---|---|---|---|---|---|---|
| | | | | 295 nm T_m | T_a | 260 nm T_m | T_a | | | |
| A1-G | CCGGGAGGAATCATGTAAGGGGTCCCGGGAACGGGCAAGGGGGCTTTTCACC | 7.0 | LiCl | nt | nt | 47.2 ± 1.2 | 40.1 ± 1.2 | 6.1 ± 0.0 | Dp | Dp |
| | | | NaCl | nt | nt | 37.8 ± 0.1 | 37.4 ± 0.2 | | | |
| | | | KCl | nt | nt | 42.8 ± 0.0 | 40.0 ± 0.3 | | | |
| | | 5.5 | KCl* | 46.2 ± 0.6 | 45.8 ± 0.6 | 47.1 ± 4.2 | 45.6 ± 0.9 | | Mix | G4 |
| A1-C | GGTGAAAAGCCCCTTGCCCGTTCCCGGGACCCCTTACACATGATTCCTCCCGG | 7.0 | LiCl | nt | nt | 12.1 ± 0.7 | 10.0 ± 0.7 | 5.7 ± 0.1 | Hp | Hp |
| | | | NaCl | nt | nt | nt | nt | | | |
| | | | KCl | nt | nt | 13.3 ± 0.9 | 12.4 ± 0.2 | | | |
| | | 5.5 | KCl*** | 39.5 ± 0.3 | 38.1 ± 0.0 | 39.0 ± 0.9 | 36.2 ± 0.1 | | iM | iM |
| A2-G | GGGTGGGGGTGATAAACGGGGCATCAGGGACACATTGCTGGGTCGGGTCGT | 7.0 | LiCl | nt | nt | 49.0 ± 0.0 | 46.1 ± 0.4 | | G4 | G4 |
| | | | NaCl | 44.1 ± 0.7 | 43.4 ± 0.4 | 44.5 ± 0.3 | 41.9 ± 0.8 | | | |
| | | | KCl | 54.2 ± 0.6 | 52.3 ± 0.8 | 40.6 ± 0.1 | 40.5 ± 0.1 | | | |
| | | 5.5 | KCl*** | 55.9 ± 0.3 | 54.3 ± 0.1 | 44.7 ± 0.0 | 44.3 ± 0.0 | | G4 | G4 |
| A2-C | ACGACCCGACCCAGCAATGTGTCCCTGATGCCCCGTTTATCACCCCCACCC | 7.0 | LiCl | 14.4 ± 0.2 | 13.4 ± 0.0 | 14.4 ± 0.6 | 10.2 ± 0.0 | 6.5 ± 0.1 | iM | iM |
| | | | NaCl | 12.4 ± 0.7 | 9.2 ± 0.1 | 11.8 ± 0.0 | 9.3 ± 0.2 | | | |
| | | | KCl | 12.5 ± 0.5 | 10.4 ± 0.2 | 13.0 ± 0.0 | 10 ± 0.0 | | | |
| | | 5.5 | KCl*** | 45.0 ± 0.8 | 42.8 ± 0.3 | 45.1 ± 0.0 | 41.5 ± 0.6 | | iM | iM |
| B-G | GCATCCCTTTTGCGGTTGTGGGGTAGGGGGCCCAGGGTAACATGAAGTTG | 7.0 | LiCl | nt | nt | nt | nt | | Dp | Dp |
| | | | NaCl | nt | nt | 39.9 ± 1.1 | 38.3 ± 0.1 | | | |
| | | | KCl | nt | nt | 40.8 ± 1.4 | 39.6 ± 0.8 | | | |
| | | 5.5 | KCl* | 46.9 ± 0.1 | 44.2 ± 0.0 | 39.0 ± 1.7 | 37.8 ± 1.7 | | Mix | G4 |
| B-C | CAACTTCATGTTACCCTGGGCCCCCCTACCCCACAACCGCAAAAGGGATGC | 7.0 | LiCl | nt | nt | 38.4 ± 0.4 | 37.9 ± 0.3 | 5.8 ± 0.3 | Hp | Hp |
| | | | NaCl | nt | nt | 37.7 ± 0.2 | 37.3 ± 0.1 | | | |
| | | | KCl | nt | nt | 33.1 ± 1.5 | 31.6 ± 0.5 | | | |
| | | 5.5 | KCl* | 36.1 ± 0.1 | 35.3 ± 0.2 | 36.5 ± 0.7 | 35.0 ± 0.0 | | iM | iM |

*iM* i-motif, *G4* G-quadruplex, *Mix* mixed species, *Hp* hairpin, *Dp* duplex, *nt* no transition observed. Cyp51 sequences and their labels (A1-G, A1-C, A2-G, A2-C, B-G and B-C. Most prominent UV melting and annealing temperatures, transitional pH (pH_T), and structure characterisation from the circular dichroism (CD) and thermal difference spectra (TDS) at indicated pH and cations. CD spectroscopy in duplicate and UV in triplicate was performed. Data from UV spectroscopy are shown as mean ± SD (n = 3). Major temperature change in bold (*P < 0.05; ***P < 0.001). Statistical significance was determined by a Welch ANOVA with Dunnett's T3 multiple comparison test. Guanines and cytosines are highlighted to facilitate identification of the G4- and iM-forming motifs, respectively.

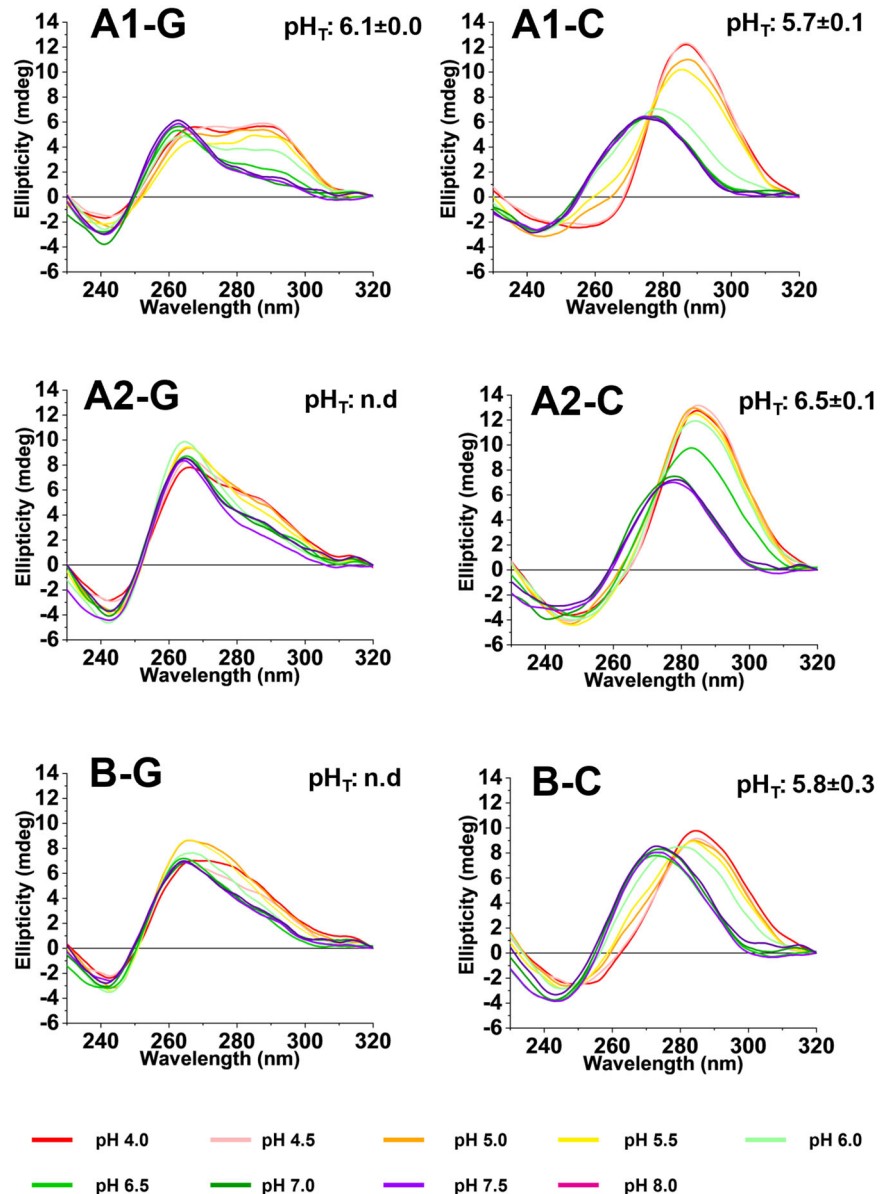

**Figure 7. Sequences from *cyp51A* and *cyp51B* can fold into G4s, iMs, hairpins/duplexes, and unusual acid-stabilised G4s.**

CD spectroscopy of the potential G4-forming oligonucleotide sequences found in the *cyp51A* coding region (A1), 3′-untranslated region (A2), and *cyp51B* coding region (B). Both C-rich and G-rich strands are shown. All spectra were obtained by annealing 10 μM DNA oligonucleotide in a buffer containing 10 mM sodium cacodylate and 100 mM KCl at pH 4–8 as indicated. The transitional pH (pH_T) was noted when a structural shift was observed and was determined from the inflection point of a Boltzmann sigmoidal. Data are representative of two replicates. Source data are available online for this figure.

Notably, PhenDC3 was equally as potent as some currently used antifungals, could synergise with amphotericin B, and was protective in a *G. mellonella* model of fungal infection. We found that G4s can potentially form in a wide range of genes implicated in the virulence and development of *A. fumigatus*. However, this is only a snapshot of some of the locations G4s may form, and we estimate that there are >1000 sequences almost guaranteed to form these structures in the *A. fumigatus* genome. Unlike humans, predicted G4s were found to be enriched within the coding regions of genes in *A. fumigatus*. Recent evidence suggests that this phenomenon is not restricted to filamentous fungi and has also been observed in *Trypanosoma brucei* and *Mycobacterium tuberculosis* (Maurizio et al, 2025; Monti et al, 2025). The former study also demonstrated that PhenDC3 specifically perturbs the expression of genes containing G4s within their coding regions, which supports our observations with *cyp51A*. This new evidence and several previous studies indicate that gene body G4s are equally important within transcriptional control and should not be overlooked when investigating their regulatory functions (Agarwal et al, 2014; Belotserkovskii et al, 2017; Holder and Hartig, 2014).

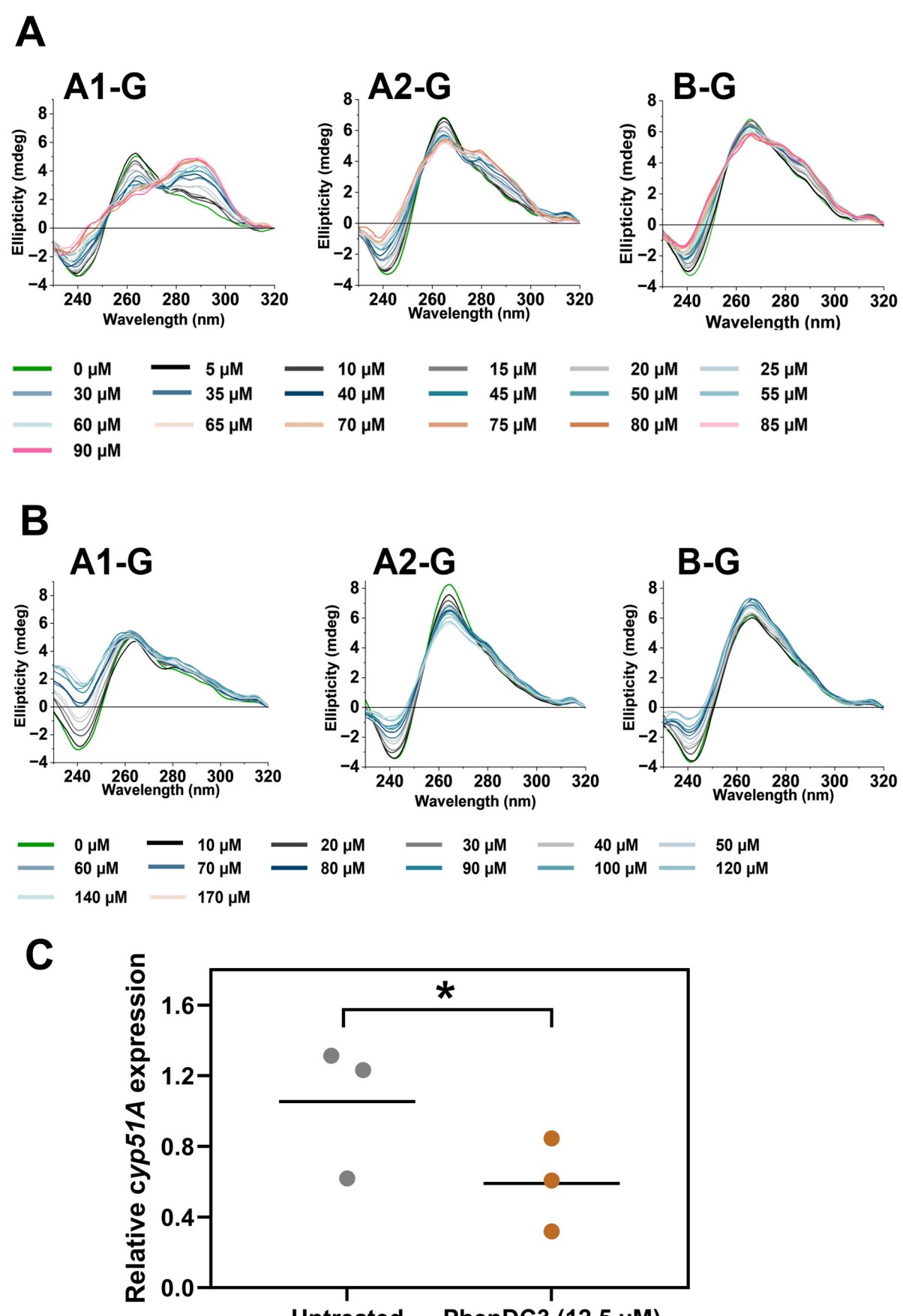

**Figure 8. PhenDC3 can cause a switch from a parallel to antiparallel G4 in *cyp51A* and reduce gene expression.**

The ellipticity of the *cyp51A* and *cyp51B* oligonucleotide sequences (10 µM) at 295 nm was determined in the presence of (**A**) 0–90 µM PhenDC3 or (**B**) 0–170 µM PDS in 10 mM sodium cacodylate and 100 mM KCl buffer at pH 7.0. Data are representative of the ellipticity at 295 nm observed in two biological replicates. Error bars represent the mean ± SD. (**C**) *cyp51A* expression was quantified by RT-qPCR. *A. fumigatus* conidia were incubated with or without 12.5 µM PhenDC3 for 4 h. RNA was extracted by TRIzol-chloroform. Primers were designed across exon-exon junctions. Cyp51A expression was normalised to TubA. Experiments were performed in biological triplicate. Each data point represents the mean of three technical replicates. Statistical significance was determined by a paired *t* test (untreated mean 1.05 ( ± 0.21 SEM), PhenDC3 treated mean 0.59 ( ± 0.15 SEM). t(4) = 4.938, P = 0.0387, (*)). Source data are available online for this figure.

Given the involvement of these non-canonical structures in regulating transcription, it is likely that PhenDC3 and PDS are downregulating the expression of these genes, as we observed for *cyp51A*. This raises the possibility of targeting G4s within fungal genomes to selectively switch off the expression of virulence genes or those involved in drug resistance. There is increased focus in the development and identification of ligands which can selectively bind to individual G4s of interest, meaning that this isn't out of the realm of possibility (Balaratnam et al, 2023; Chowdhury et al, 2022; Tassinari et al, 2020).

We are not currently sure why PhenDC3 and amphotericin B synergise, but it may be that this interaction is similar to that observed for flucytosine, and the increased permeability of the cell membrane allows more PhenDC3 to enter the fungi. However, we do not observe any synergy between amphotericin B and PDS, which means that this interaction may be more complex. Further, the G4-binding natural product, berberine, has previously been shown to inhibit ergosterol synthesis (Lei et al, 2011). This mechanism is shared with the azoles, which may explain why PhenDC3 cannot synergise with azoles, but this hypothesis is currently quite speculative.

Although both PhenDC3 and PDS are designed to bind to G4s, their antifungal mechanism of action was very different. The way in which these ligands interact with DNA might provide some insight. We found that PhenDC3 caused the refolding of the A1-G and A2-G sequences into antiparallel G4 structures. Indeed, PhenDC3 has previously been shown to induce refolding of a hybrid G4 in the telomeres to an antiparallel chair-type structure by intercalating into the DNA (Ghosh et al, 2022). This is important to note as parallel G4s act as a substrate for telomerase binding, but antiparallel or hybrid G4s are poorly utilised as substrates for telomerase (El-Khoury et al, 2023; Paudel et al, 2020; Zahler et al, 1991; Zaug et al, 2005). It is possible that PhenDC3's capability of inducing the non-native antiparallel form of the G4 prevents protein binding and inhibits essential pathways in fungi. There were significant differences between the antifungal potencies of the G4-ligands between different species. Considering we have previously shown that both *A. flavus* and *A. niger* have higher frequencies of G4-forming sequences and equivalent GC genome content to *A. fumigatus*, the loss of PhenDC3 activity is difficult to explain (Warner et al, 2021). However, this is suggestive that G4-ligands may be preventing the function of a specific gene (or set of genes) critical for the survival of these fungi. Indeed, through our analyses, we have found G4s to be present in some essential genes within *A. fumigatus*, but we need to explore this further. Additionally, these species may have more active drug efflux

transporters, and *A. flavus* has been shown to have a very different cell wall to *A. fumigatus* which contributes to increased resistance to cell wall-targeting drugs; meaning penetration could be an issue (Wong et al, 2021). Increased drug efflux and drug penetration issues may also underlie why PhenDC3 was ineffective against mature *A. fumigatus* biofilms, as this has been observed previously for other antifungals (Rajendran et al, 2011). It has also previously been observed that antifungal drugs such as the azoles and amphotericin B are 1000× less effective against *A. fumigatus* biofilms. Therefore, it may be likely that higher concentrations of PhenDC3 would be antifungal, but the increased risk of toxicity does not make this a viable therapeutic option (Mowat et al, 2007).

Alternatively, accessibility to the G4s may differ between species as chromatin structure has a strong influence on G4 formation. Most endogenous G4s can be found mapped to nucleosome-depleted, open regions of chromatin, and certain G4s may be inaccessible to G4-ligands across species (Shen et al, 2021). Although this doesn't explain the faster and broader activity of PDS. It may be that PDS exerts its antifungal activity non-specifically and could simply be due to the promotion of genome instability and inhibition of DNA replication as observed in *S. cerevisiae* (Lopes et al, 2011; Paeschke et al, 2013; Piazza et al, 2010).

Unusually, we also identified two G4s in *cyp51A* and *cyp51B*, which transitioned to stable mixed species G4s at pH 5.5. It is widely appreciated that the cytosine-rich iM is readily stabilised by acidic pH, but acidity does not typically stabilise G4s (Day et al, 2014; König et al, 2013). There has been at least one previously reported case of an acid-stabilised G4 that can form within the telomeric region, but reports are rare (Galer et al, 2016). We are currently unsure why this occurs in these *cyp* genes and what implications this might have on the function of both Cyp51A and Cyp51B, but both proteins are implicated in azole resistance and acidity has been shown to affect azole resistance in fungi (Lourenço et al, 2018; Song et al, 2022). Therefore, the formation of the mixed G4 in acidic conditions might be a contributing factor to azole resistance and should be explored further.

These findings offer the first insights into the potential therapeutic applications of stabilising distinct secondary structures within the genomes of fungal pathogens, with further implications for genomic stability, drug resistance, and regulation in *Aspergillus* and *Candida* species. However, further study is required to identify the specific targets of G4-stabilisers and to elucidate the biological roles of G4s in fungi to help develop novel antifungals targeting specific G4s.

# Methods

### Reagents and tools table

| Reagent/resource | Reference or source | Identifier or catalogue number |
|---|---|---|
| **Experimental models** | | |
| A549 epithelial cells | Dr Anastasia Sobolewski | |
| Vascular smooth muscle cells | Cell Applications Inc. USA | 354-05a |
| *Galleria mellonella* | Live Foods Direct, UK | |
| *Aspergillus fumigatus* | This study | 22M7007854 |
| *Aspergillus fumigatus* | This study | 22M7004177 |
| *Aspergillus fumigatus* | https://doi.org/10.1128/jcm.31.6 | CEA10 |
| *Aspergillus fumigatus* | https://doi.org/10.1093/jac/dkt075 | A1160+ |
| *Aspergillus fumigatus* | https://doi.org/10.1016/j.jgar | TR$_{34}$/L98H |
| *Aspergillus hiratsukae* | This study | 22M7007855 |
| *Aspergillus udagawae* | This study | 22M7007856 |
| *Aspergillus flavus* | This study | 22M8001325 |
| *Aspergillus brasiliensis* | https://doi.org/10.1099/ijs.0.65021-0 | ATCC16404 |
| *Aspergillus niger* | This study | 22M8001245 |
| *Candida albicans* | https://doi.org/10.1128/MCB.00313-10 | JC747 |
| *Candida auris* | https://doi.org/10.1093/mmy/myw147 | NCPF8985 |
| *Candida glabrata* | https://doi.org/10.1038/nature0259 | CBS-138 |
| **Antibodies** | | |
| Phospho-Histone H2A.X (Ser139) Antibody | Cell Signalling Technology | 2577 |
| Goat anti-mouse IgG (H + L) Alexa Fluor 488 secondary antibody | Invitrogen | A-11001 |
| **Oligonucleotides and other sequence-based reagents** | | |
| rodA | **GGGG**TA**GGG**AACA**GGG**TCCC**GGG** | Merck |
| arp2 | **GGGG**A**GGGG**ACCCTTGCAT**GGAGGGGGGG** | Merck |
| ags1 | **GGG**CTTGA**GGGG**AACT**GGG**AGGAGC**GGG** | Merck |
| cyp51A | **GGG**T**GGGGG**TGATAAAC**GGGG**CATCA**GGG** | Merck |
| cyp51B | **GGGG**TA**GGGGGG**CCCA**GGG** | Merck |
| hmg1 | **GGG**C**GG**CA**GGG**ACA**GGGG**CAACTTC**GGG** | Merck |
| atrR | **GGGGG**A**GGG**TGC**GGG**ATGCTCTTC**GGG** | Merck |
| erg1 | **GGG**CGAC**GGGG**AAGAGGTA**GGG**A**GGG** | Merck |
| erg5 | **GGG**ATTA**GGG**TATG**CCTCA**GGG**TCAT**GGG** | Merck |
| mepB | **GGG**CGA**GGG**CAGT**GGG**TTCGT**GGG** | Merck |
| dppV | **GGG**AC**GG**CGTT**GGGGGGGGGG**AA**GGG** | Merck |
| mirB | **GGG**TAGCTGAGCT**GGG**T**GGGG**A**GGG** | Merck |
| zrfC | **GGG**TT**GGG**CTGGTCGACATGT**GGG**CCC**GGG** | Merck |
| zafA | **GGG**CA**GGG**CAGAGCA**GGG**CA**GGGG** | Merck |
| lysF | **GGG**ACGC**GGGGGGG**C**GG**CATC**GGG**AGC**GGG** | Merck |
| hscA | **GGGG**CTGAATT**GGGGGGGG**AAAAC**GAGAC**GGG** | Merck |
| aspf2 | **GGGG**AC**GGGGG**A**GG**TA**GGG**A**GGG** | Merck |
| aspf28 | **GGG**TTGCGA**GGG**AAG**GAGGGG**C**GGG** | Merck |
| gstA | **GGG**CT**GGG**TGGC**GG**CCGCT**GG**TT**GGG**C**GGG** | Merck |

| Reagent/resource | Reference or source | Identifier or catalogue number |
|---|---|---|
| *yap1* | **GGG**AT**GGG**CTTCCCAAGTC**GGGG**AGA**GGG** | Merck |
| *skn7* | **GGGGGGGGGG**T**GGG** | Merck |
| *pes1* | **GGG**AGCAAG**GGG**AAATCGCTGA**GGG**T**GGG** | Merck |
| *atrF* | **GGG**TATTT**GTTTGGGGGGG**AT**GGG** | Merck |
| *abcB* | **GGGGG**T**GGG**CGTT**GGGG**C**GGG** | Merck |
| *fmpD* | **GGGGGG**ACCCGTCAT**GGG**C**GGG** | Merck |
| *mfsB* | **GGG**AGACAA**GGGGG**A**GGGGGG** | Merck |
| *mnt1* | **GGG**AGA**GGG**CACA**GGGGGGG** | Merck |
| *hsp70* | **GGG**ACACCAC**GGGGGGG**C**GGG** | Merck |
| *pacC* | **GGGGGGGGGG**AATGCTT**GGG** | Merck |
| *ste7* | **GGG**AG**GGG**ACACAAAGAG**GGGG**ATAA**GG**AG**GGG** | Merck |
| *bck1* | **GGG**CCTGGAG**GGGGGG**CCC**GGGG** | Merck |
| *pkaC1* | **GGG**TTGT**GGT**G**GGT**G**G**AG**GG**AT**GGGG** | Merck |
| *laeA* | **GGG**ATGT**GGG**ACA**GGG**ATTT**GGG** | Merck |
| *gliN* | **GGGGGG**CGCTT**GG**CT**GGG**AAGGC**GG**TT**GGG** | Merck |
| *ftmC* | **GGG**TCT**GG**AGCGC**GGGG**TTC**GGGG**ATC**GGG** | Merck |
| *fks1* | **GG**AA**GGGG**CTC**GGG**CAT**GGG** | Merck |
| *mp1* | **GG**CTCC**GG**CTCC**GG**CTCC**GG** | Merck |
| *chsG* | **GG**AGAGC**GGG**AG**G**T**GGT**GA**GGG** | Merck |
| *ftmE* | **GGG**AGT**GGGG**ATGGCGAC**GGGG**AGCA**GGG** | Merck |
| B DNA | **GG**CGCTATTCTCT**GGT**GCT**G**CT | Merck |
| Telomere | TTA**GGG**TTA**GGG**TTA**GGG**TTA**GGG**TT | Merck |
| A1-G | CC**GGG**A**GG**AATCAT**G**TAA**GGGG**TCCC**GGG**AAC**GGG**CAA**GGGG**CTTTTCACC | Eurogentec |
| A1-C | GGTGAAAAG**CCCC**TTG**CCC**GTT**CCC**GGGA**CCCC**TTACATGATT**CCT**C**CCC**GG | Eurogentec |
| A2-G | **GGG**T**GGGGG**TGATAAAC**GGGG**CATCA**GGG**ACACATTGCT**GGG**TC**GGG**TCGT | Eurogentec |
| A2-C | AC**G**AC**CCG**AC**CC**AGCAATGTGT**CCC**TGATG**CCCC**GTTTAT**CACCCCC**A**CCC** | Eurogentec |
| B-G | **G**CATC**CC**TTTT**G**C**GG**TT**G**T**GGGG**TA**GGGGGG**CCCA**GGG**TAACAT**G**AA**G**TTG | Eurogentec |
| B-C | **C**AA**C**TT**C**ATGTTA**CCC**T**GGGCCCCCC**TA**CCCC**ACAAC**C**GCAAAAG**GG**ATG**C** | Eurogentec |
| Cyp51a_F | TGCGTGCAGAGAAAAGTATGG | Merck |
| Cyp51a_R | AGACCTCTTCCGCATTGACA | Merck |
| TubA_F | CTCCGTTCCTGAGTTGACCC | Merck |
| TubA_R | CACGGAAAATGGCAGAGCAG | Merck |
| **Chemicals, enzymes and other reagents** | | |
| Sabouraud dextrose agar | Oxoid | CM0041B |
| PBS | Oxoid | OXBR0014G |
| Tween-20 | MP Biomedicals | 500-018-3 |
| DMEM F-12 | Gibco/ Thermo Fisher Scientific | 11320033 |
| Foetal bovine serum | Thermo Scientific | A5669201 |
| Penicillin/Streptomycin | Gibco/Thermo Fisher Scientific | 15070063 |
| Trypsin | Gibco/Thermo Fisher Scientific | 25200072 |
| Amphotericin B | Thermo Fisher Scientific | 15290026 |
| PhenDC3 | Merck | SML2298 |
| Pyridostatin (PDS) | Merck | SML2690 |
| RPMI-1640 | Gibco/ Thermo Fisher Scientific | 11875093 |

| Reagent/resource | Reference or source | Identifier or catalogue number |
| --- | --- | --- |
| Smooth muscle cell media | Cell Applications Inc. USA | 311-500 |
| MOPS | Thermo Fisher Scientific | BP308100 |
| Resazurin salt | Merck | R7017 |
| Itraconazole | Merck | I6657 |
| Voriconazole | Merck | PZ0005 |
| Glutaraldehyde | Merck | 354400 |
| Sodium cacodylate | Thermo Fisher Scientific | AAJ60344AD |
| Osmium tetroxide | Thermo Fisher Scientific | AC319010100 |
| Poly-L-Lysine | Thermo Fisher Scientific | A005C |
| Hoechst 33258 | Merck | 94403 |
| QUMA-1 | Merck | SCT056 |
| TRIzol | Invitrogen | 15596026 |
| Chloroform | Merck | 319988 |
| Isopropanol | Merck | 190764 |
| Ethanol | Merck | 493546 |
| NP-40 | Thermo Fisher Scientific | J60766.AP |
| Bovine serum albumin | Merck | A8806 |
| Rhodamine phalloidin | Invitrogen | R415 |
| VECTASHIELD Antifade mounting medium with DAPI | Vector Laboratories | H-1200-10 |
| Thioflavin T | Merck | 596200 |
| LiCl | Merck | L7026 |
| NaCl | Thermo Fisher Scientific | BP358-212 |
| KCl | Thermo Fisher Scientific | 11-101-5415 |
| **Software** | | |
| GraphPad Prism Version 10.0.3 | GraphPad Software | |
| OriginPro Version 2023B | OriginLab Corporation | |
| **Other** | | |

## Fungal isolates and culture

*Aspergillus* spp. and *Candida* spp. (Appendix Table S1) were routinely grown on Sabouraud Dextrose Agar (SDA; Oxoid, UK) for 3–5 days for *Aspergillus* spp. and 1–2 days for *Candida* spp. at 37 °C in T-75 tissue culture flasks (Thermo Fisher Scientific, UK). *Aspergillus* conidia were harvested using PBS-Tween-20 (0.05%), and large hyphal fragments were removed by passing the harvested fungi through Miracloth (VWR Scientific, USA). Conidia were pelleted at $3000\times g$ for 5 min and washed twice in sterile PBS. *Candida* was harvested by inoculating 3–5 distinct colonies in sterile water.

## Tissue culture

The type II alveolar epithelial/adenocarcinoma cell line, A549 (a kind gift from Dr Anastasia Sobolewski, University of East Anglia) was maintained in DMEM F-12 supplemented with 2 mM L-glutamine (Gibco, Thermo Fisher Scientific, UK), 10% foetal bovine serum (Thermo Fisher Scientific, UK), and 100 U/mL penicillin plus 100 µg/mL streptomycin (Thermo Fisher Scientific, UK). Primary vascular smooth muscle cells were purchased from Cell Applications Inc. (USA; Cat#354-05a). Standard culture (passages 3–9) was performed as previously described (Ragnauth et al, 2010; Warren et al, 2015). Cells were maintained at 37 °C in a humidified incubator supplied with 5% $CO_2$. Cell lines were tested for mycoplasma contamination at source.

## Antifungal susceptibility testing

Antifungal susceptibility assays were conducted according to adapted Clinical and Laboratory Standards Institute (CLSI) reference methods: M27Ed4 for *Candida* spp. and M38Ed3 for *Aspergillus* spp. Conidia or yeast were incubated with DMSO as a vehicle control, amphotericin B (2 µg/mL; Thermo Fisher Scientific, UK) as a positive control, or the G4-ligands PhenDC3 and PDS (both purchased from Merck, UK) in RPMI-1640 growth medium (Gibco, UK) buffered with 0.165 M MOPS (3-(N-morpholino) propanesulfonic acid) buffer (pH 7.0) at 37 °C for 24 h or 48 h where indicated. The final concentration of DMSO in each instance

was <0.05%. Following incubation, 10 µg/mL resazurin (Merck, UK) in PBS was added to each well and incubated for 4 h for *Candida* spp. or 24 h for *Aspergillus* spp. at 37 °C. The ability of metabolically active cells to convert resazurin to the fluorescent product resorufin was quantified with a CLARIOstar plate reader (BMG Labtech, UK) using an excitation wavelength of 530 nm and an emission wavelength of 590 nm.

For checkerboard assays, *Aspergillus* spp. Conidia ($1 \times 10^6$) were added to each well of a 96-well plate containing RPMI-1640 growth medium buffered with 0.165 M MOPS (pH 7.0). PhenDC3 (0.09–6.25 µM), amphotericin B, itraconazole, or voriconazole (all at concentrations from 0.031 to 8 µg/mL and purchased from Merck or Thermo Fisher Scientific, UK) were added to conidia for 24 h alone or in combination at 37 °C. Metabolic activity was determined as above following incubation with 10 µg/mL resazurin for 24 h. Concentrations exhibiting a >50% decrease in viability was used as a cut-off for activity in the checkerboard assays, and the fractional inhibitory concentration index was determined using the following equation:

$$\frac{A}{MIC_A} + \frac{B}{MIC_B} = FIC_A + FIC_B = FIC\ index\ value$$

In all instances, the background fluorescence after amphotericin B treatment was subtracted from the test samples, as this value represented the intrinsic fluorescence of the dye in the absence of viable fungi (0% viability). Metabolic activity was calculated as a percentage of the untreated control (100% viability).

## Scanning electron microscopy (SEM)

Conidia ($1 \times 10^6$) were treated with 12.5 µM PhenDC3 or PDS for 4 h at 37 °C. Following incubation, conidia were pelleted in PBS containing 0.05% Tween-20 at $3000 \times g$ for 5 min and fixed in 2.5% glutaraldehyde for 3 h. Fixed conidia were washed and resuspended in 0.05 M sodium cacodylate buffer (pH 7.3), transferred onto 13 mm poly-L-lysine coated coverslips (0.01% w/v), and left to settle for 20 min. Conidia were post-fixed in 1% (v/v) $OsO_4$ for 1 h at room temperature. The $OsO_4$ fixation was followed by three 15 min washes in distilled water before beginning the ethanol dehydration series (25%, 50%, 75%, 95% and two changes of 100% ethanol, each for an hour) and left to air dry overnight. The sample was then plunged into a liquid nitrogen slush and transferred under vacuum onto a PP3010T Cryo-SEM preparation system (Quorum Technologies, UK) at −140 °C. The sample was sublimated for 2 min at −90 °C before cooling again to −140 °C and sputter-coating with platinum for 1 min at 10 mA. The sample was introduced onto the cryo-stage in a Gemini 300 scanning electron microscope (Carl Zeiss, Germany) and viewed at 2 kV.

## *Galleria mellonella* infection model

Larvae of *Galleria mellonella* (Live Foods Direct, UK) were used for all experiments. Larvae were maintained in their box in the dark at room temperature prior to use. Only larvae weighing between 100 and 300 mg, with normal mobility, and uniform colour were selected for the experiments. Larvae were randomly distributed into groups of 10. Larvae were administered $1 \times 10^6$ *A. fumigatus* conidia in sterile PBS via injection in the ventral side of the last proleg using a Hamilton® syringe. Groups of larvae injected with

PBS alone, 2.5 mg/kg PhenDC3 alone, 2.5 mg/kg amphotericin B alone, or not injected at all were included as controls. Two hours post-infection, either 2.5 mg/kg amphotericin B, 0.25 mg/kg PhenDC3, or a combination of 0.25 mg/kg PhenDC3 and 0.25 mg/kg amphotericin B were administered via injection into the haemocoel. Infected larvae were stored in the dark at 37 °C, and mortality was monitored for 7 days. Larvae were considered dead when they melanised and stopped responding to stimulation. All treatments were prepared in sterile PBS and administered in a 10 µL inoculum. Experiments were performed in triplicate and results were pooled.

## Cytotoxicity assays

To assess the cytotoxicity of PhenDC3 and PDS against A549 cells and primary vascular smooth muscle cells, cells were plated at a density of $2 \times 10^4$ cells/well in their respective growth buffers in a 96-well plate and incubated at 37 °C and 5% $CO_2$ for 24 h. The following day, the growth media was aspirated and replaced with growth media containing PhenDC3 or PDS at a concentration range from 0.7 to 100 µM. Cells were incubated for a further 24 h prior to the addition of resazurin (10 µg/mL) and incubation at 37 °C and 5% $CO_2$ for 2 h. The ability of metabolically active cells to convert resazurin to the fluorescent product resorufin was quantified with a CLARIOstar plate reader using an excitation wavelength of 530 nm and an emission wavelength of 590 nm.

## Determination of PQS frequency and location in *Aspergillus fumigatus*

The G4Hunter script (https://github.com/LacroixLaurent/G4Hunter) was used to identify putative G4s as described previously (Bedrat et al, 2016). When analysing the genomes, the scored nucleic acid sequence was computed for a sliding window of 25 nucleotides and a threshold above 1.2, which highlights sequences which are most likely to form secondary structures.

## QUMA-1 assay

Fungi were treated with 12.5 µM of the G4-ligands PhenDC3 or PDS in a glass-bottomed µ-Plate 96-well black plate (Ibidi, Germany) for 18 h prior to labelling with QUMA-1 (Merck, UK). Following treatment, fungi were washed in PBS + 0.05% Tween-20 and resuspended in PBS containing 1 µM Hoechst 33342 (Merck, UK) and 1 µM QUMA-1. Live fungi were stained at 37 °C for 1 h and imaged immediately on a Zeiss LSM 980 laser scanning confocal microscope equipped with an Airyscan 2 detector (Carl Zeiss, Germany) using a ×100 oil objective. The average foci per 10 µm² in all fungi from five random images per treatment from three independent experiments were quantified using Fiji 1.53t.

## RT-qPCR

Total RNA was extracted from cells using TRIzol reagent (Invitrogen, UK) with chloroform separation and isopropanol precipitation. RNA was quantified using Luna® Universal One-Step RT-qPCR Kit (New England Biolabs, UK). Briefly, 1 µg of isolated total RNA, oligonucleotides (Cyp51a_F, Cyp51a_R, TubA_F, TubA_R), and Luna® Reaction Mix were added to a final volume

of 20 μl. Reverse transcription was carried out at 55 °C for 10 min, followed by the initial denaturation at 95 °C for 1 min. Then, 40 thermocycles at 95 °C for 1 min and 60 °C for 30 s were performed. PCR quantification was conducted using the $2 - \Delta\Delta CT$ method and normalised to TubA expression.

The primers used were as follows:

Cyp51a_F - TGCGTGCAGAGAAAAGTATGG
Cyp51a_R - AGACCTCTTCCGCATTGACA
TubA_F - CTCCGTTCCTGAGTTGACCC
TubA_R - CACGGAAAATGGCAGAGCAG.

## DNA damage

Vascular smooth muscle cells were seeded in 6-well plates in VSMC growth media for 24 h at 37 °C and 5% $CO_2$. Following incubation, the growth media was aspirated and replaced with growth media containing 25 μM or 50 μM PhenDC3/PDS and incubated for 24 h at 37 °C and 5% $CO_2$. The media was removed, and cells were washed with PBS prior to fixation with paraformaldehyde for 10 min and permeabilization with 0.5% NP-40 (Thermo Fisher Scientific, UK) for 5 min. Cells were blocked with 3% bovine serum albumin (Merck, UK) for 1 h and stained with a γH2AX primary antibody (1:100; Cell Signalling Technology, UK) overnight at 4 °C. The primary antibody was washed away with PBS prior to the addition of the Alexa Fluor anti-mouse 488 secondary antibody (1:400; Invitrogen, UK) and rhodamine phalloidin stain (1:400; Invitrogen, UK). Cells were incubated in the dark at room temperature for 2 h prior to washing three times with PBS and mounting on microscope slides using DAPI mounting media (Vector Laboratories, UK). Images were obtained using a Zeiss LSM 980 laser scanning confocal microscope equipped with an Airyscan 2 detector (Carl Zeiss, Germany) using the 20x objective. Images were analysed using Fiji 1.53t.

## Oligonucleotides

The DNA sequences used for biophysical characterisation were synthesised and reverse-phase high-performance liquid chromatography (HPLC)-purified by Eurogentec (Eurogentec, Belgium). The DNA sequences used for the thioflavin T assay were purchased from Merck. The oligonucleotides were dissolved in ultra-pure water to a final concentration of 1 mM, and their concentrations were confirmed using a NanoDrop (Thermo Fisher Scientific, UK). Sodium cacodylate was used in line with previous studies at 10 mM with 100 mM of either LiCl, NaCl, or KCl at the specified pH (Martella et al, 2022). For biophysical analysis, samples were annealed by heating at 95 °C for 5 min in a heating block, followed by cooling to room temperature overnight.

## Thioflavin T assay

Oligonucleotides (4 μM final concentration) were annealed at 95 °C for 5 min in 10 mM sodium cacodylate buffer, 100 mM KCl, pH 7.4, and cooled to room temperature for 2 h. The annealed oligonucleotides were transferred to a black, flat-bottomed, 96-well plate (Nunc, Denmark), and 1 μM thioflavin T (Merck, UK) was added to each of the oligonucleotide-containing wells. Fluorescence intensity was quantified on a CLARIOstar plate reader with an excitation wavelength of 425 nm and an emission spectrum from 450 to 700 nm.

## Circular dichroism (CD) spectroscopy

The CD spectra of selected fungal *cyp51* sequences at various pH values were recorded using a JASCO J-1500 spectropolarimeter under a constant flow of nitrogen gas (JASCO, UK). Samples were diluted to 10 μM in 10 mM sodium cacodylate buffer containing 100 mM LiCl or 100 mM NaCl at pH 7, and 100 mM KCl buffer at pH values ranging from 4 to 8. Four scans were accumulated for each sample and corresponding buffer blank over a wavelength range of 230–320 nm at 25 °C. The instrument settings were as follows: data pitch of 0.5 nm, scanning speed of 200 nm/min, response time of 1 s, bandwidth of 1 nm, and sensitivity of 200 mdeg. Data were zero-corrected at 320 nm. The transitional pH ($pH_T$) was determined from at least two independent experiments by identifying the inflection point of the sigmoidal fit of ellipticity measured at 288 nm or 295 nm, as appropriate for the sequence and pH range.

CD spectroscopy was also used to measure any ligand-induced effects on the fungal *cyp51* sequences (A1-G, A2-G and B-G). All samples were diluted to 10 μM in 10 mM sodium cacodylate buffer containing 100 mM KCl at pH 7.0. Four scans were accumulated over the wavelength range of 230–320 nm at 25 °C with the same instrument settings as above. Ligands PhenDC3 and PDS were titrated stepwise into the DNA samples via consecutive additions, with final ligand concentration ranges of 0–90 μM for PhenDC3 and 0–170 μM for PDS. Titrations were performed in duplicate, and data were corrected for solvent effects, smoothed using the Savitzky–Golay method with a 10-point window, and zero-corrected at 320 nm. Data analysis was conducted using OriginLab software (version 2023B; OriginLab Corporation, USA). The sigmoidal curves of ellipticity at 295 nm versus ligand concentration were fitted to the Hill equation to obtain the Hill coefficient (n) and K, representing the ligand concentration required to achieve 50% reduction of molar ellipticity.

## UV melting/annealing and thermal difference spectroscopy (TDS)

UV melting/annealing and TDS experiments were performed using a Jasco V-750 UV–Vis spectrophotometer (JASCO, UK). Samples were prepared at a concentration of 5 μM in 10 mM sodium cacodylate buffer containing 100 mM LiCl, 100 mM NaCl, or 100 mM KCl at pH 7, and 100 mM KCl at pH 5.5. For the UV melting and annealing experiments, absorbance was recorded at 260 nm and 295 nm every 1 °C over three thermal cycles. Samples were initially held at 4 °C for 10 min, followed by a gradual increase to 95 °C at a rate of 0.5 °C/min (melting phase). Upon reaching 95 °C, samples were held for 10 min before reversing the process to return to 4 °C (annealing phase). At each temperature interval, samples were equilibrated for 5 min before measurement. The most prominent melting temperature ($T_m$) and annealing temperature ($T_A$) were determined using the first derivative method for each cycle.

Thermal difference spectra were obtained by measuring the absorbance spectra from 230–320 nm after equilibrating the samples at 4 °C for 10 min (folded DNA) and at 95 °C for 10 min (unfolded DNA). The TDS signatures were calculated by subtracting the absorbance spectra of the folded structures from those of

**The paper explained**

**Problem**

Fungal infections are responsible for millions of deaths each year, and the problem is expected to worsen with the rise of antifungal resistance. This highlights an urgent need for new antifungal agents with novel mechanisms of action.

**Results**

The G-quadruplex (G4) ligands PhenDC3 and pyridostatin demonstrated both fungistatic and fungicidal activity against azole-susceptible and resistant fungi. PhenDC3 was more potent and could inhibit germination of *A. fumigatus* whilst also synergising with the antifungal amphotericin B. G4s were found to form in the RNA of *A. fumigatus* and their formation was promoted by PhenDC3. G4-forming sequences were identified in *cyp51A* and *cyp51B*, genes which play a crucial role in azole resistance. Notably, PhenDC3 could cause the G4 in *cyp51A* to change shape, and this change was associated with downregulated expression of *cyp51A*. Further, a rare instance of acid-stabilised G4-formation was observed for the sequences identified in *cyp51A*.

**Impact**

G4s may represent novel regulators of fungal biological processes and act as targets for the development of new antifungals.

the unfolded structures, zero-corrected at 320 nm, and normalised to the maximum absorbance.

## Statistics

The sample sizes ($n$), the probability ($P$) value, and the statistical test used to analyse each experiment can be found in the figure legends. Statistical significance was determined by paired Student's *t* test to compare two paired groups. To compare multiple groups, a Kruskal–Wallis with Dunn's multiple comparisons or Brown-Forsythe and Welch ANOVA with Dunnett's T3 multiple comparisons was conducted. Survival analyses were performed using Kaplan–Meier survival plots. Normality was determined via the Shapiro–Wilk test. Data was prepared using GraphPad Prism (version 10.0.3) and OriginPro (version 2023B). Data from most experiments are presented as mean values ± SEM from three independent experiments. Data from UV spectroscopy are shown as mean ± SD. $P < 0.05$ was considered statistically significant. Data are representative of at least two independent experiments. GraphPad Prism was unable to provide exact $P$ values for Figs. 3D and 6B due to the post-hoc tests used and sample sizes. In these instances, the approximate $P$ values are provided.

## Data availability

The source data of this paper are collected in the following database record https://doi.org/10.5281/zenodo.17192248.

The source data of this paper are collected in the following database record: biostudies:S-SCDT-10_1038-S44321-025-00340-1.

## Peer review information

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

## Acknowledgements

We are thankful for the support of A Sobolewski (University of East Anglia, Norwich) with the A549 cells used in this study. This work was supported by the Academy of Medical Sciences [SBF008\1046 to SB] and [SBF0010\1140 to TF], British Mycological Society to TF, Biotechnology and Biological Sciences Research Council [BB/Y005058/1 to SB; BB/W001616/1 to ZAEW], Wellcome Trust [226408/Z/22/Z to NvR], Royal Society Research Grant [RG \R2\232363] and the NIHR Newcastle Biomedical Research Centre (BRC) to AdSD, and Thomas Marns Scholarship to FOM. The funders had no role in study design, data collection and analysis, decision to publish, or preparation of the manuscript.

## Author contributions

**Georgie Middleton**: Data curation; Formal analysis; Investigation; Visualisation; Methodology; Writing—original draft; Writing—review and editing. **Fuad O Mahamud**: Data curation; Formal analysis; Investigation; Visualisation; Methodology; Writing—original draft; Writing—review and editing. **Isabelle S R Storer**: Data curation; Formal analysis; Supervision; Investigation; Visualisation; Methodology; Writing—original draft; Writing—review and editing. **Abigail Williams-Gunn**: Data curation; Formal analysis; Investigation; Writing—review and editing. **Finn Wostear**: Formal analysis; Investigation; Methodology; Writing—review and editing. **Alireza Abdolrasouli**: Resources; Investigation; Writing—review and editing. **Elaine Barclay**: Resources; Investigation; Writing—review and editing. **Alice Bradford**: Investigation; Methodology; Writing—review and editing. **Oliver Steward**: Investigation; Methodology; Writing—review and editing. **Silke Schelenz**: Resources; Investigation; Writing—review and editing. **James McColl**: Investigation; Methodology; Writing—review and editing. **Betrand Lézé**: Investigation; Methodology; Writing—review and editing. **Norman van Rhijn**: Funding acquisition; Investigation; Writing—review and editing. **Alessandra da Silva Dantas**: Investigation; Writing—review and editing. **Takanori Furukawa**: Investigation; Writing—review and editing. **Derek Warren**: Conceptualisation; Supervision; Writing—review and editing. **Zoë A E Waller**: Conceptualisation; Supervision; Funding acquisition; Writing—original draft; Project administration; Writing—review and editing. **Stefan Bidula**: Conceptualization; Formal analysis; Supervision; Funding acquisition; Writing—original draft; Project administration; Writing—review and editing.

Source data underlying figure panels in this paper may have individual authorship assigned. Where available, figure panel/source data authorship is listed in the following database record: biostudies:S-SCDT-10_1038-S44321-025-00340-1.

## Disclosure and competing interests statement

NvR has competing interests and has received speaker honoraria from Napp/Mundipharma. The remaining authors declare no competing interests.

