## [Peer Review File · EMBO Molecular Medicine]

Evidence that G-quadruplexes form in pathogenic fungi and represent promising antifungal targets

Georgie Middleton, Fuad Mahamud, Isabelle Storer, Abigail Williams-Gunn, Finn Wostear, Alireza Abdolrasouli, Elaine Barclay, Alice Bradford, Oliver Steward, Silke Schelenz, James McColl, Bertrand Lézé, Norman van Rhijn, Alessandra da Silva Dantas, Takanori Furukawa, Derek Warren, Zoë Waller, and Stefan Bidula

Corresponding authors: Stefan Bidula (s.bidula@uea.ac.uk) , Zoë Waller (z.waller@ucl.ac.uk)

Review Timeline:

Submission Date:	3rd Jun 25
Editorial Decision:	30th Jun 25
Revision Received:	6th Oct 25
Editorial Decision:	14th Oct 25
Revision Received:	23rd Oct 25
Accepted:	29th Oct 25

Editor: Zeljko Durdevic

Transaction Report:

30th Jun 2025

Dear Dr. Bidula,

Thank you for the submission of your manuscript to EMBO Molecular Medicine. We have now received feedback from the three reviewers who agreed to evaluate your manuscript. As you will see from the reports, all three referees recognize potential interest of the study, but they also raise important concerns that should be addressed in a major revision. If you would like to discuss further the points raised by the referees, I am available to do so via email or video. Let me know if you are interested in this option.

We would welcome the submission of a revised version within three months for further consideration. Please let us know if you require longer to complete the revision.

I look forward to receiving your revised manuscript.

Yours sincerely,

Zeljko Durdevic

Zeljko Durdevic
Senior Editor
EMBO Molecular Medicine

We require:

- 1) A .docx formatted version of the manuscript text (including legends for main figures, EV figures and tables). Please make sure that the changes are highlighted to be clearly visible.
- 2) Individual production quality figure files as .eps, .tif, .jpg (one file per figure). For guidance, download the 'Figure Guide PDF': (<https://www.embopress.org/page/journal/17574684/authorguide#figureformat>).
- 3) A .docx formatted letter INCLUDING the reviewers' reports and your detailed point-by-point responses to their comments. As part of the EMBO Press transparent editorial process, the point-by-point response is part of the Review Process File (RPF), which will be published alongside your paper.
- 4) A complete author checklist, which you can download from our author guidelines (<https://www.embopress.org/page/journal/17574684/authorguide#submissionofrevisions>). Please insert information in the checklist that is also reflected in the manuscript. The completed author checklist will also be part of the RPF.
- 5) Please note that all corresponding authors are required to supply an ORCID ID for their name upon submission of a revised manuscript.
- 6) It is mandatory to include a 'Data Availability' section after the Materials and Methods. Before submitting your revision, primary

datasets produced in this study need to be deposited in an appropriate public database, and the accession numbers and database listed under 'Data Availability'. Please remember to provide a reviewer password if the datasets are not yet public (see <https://www.embopress.org/page/journal/17574684/authorguide#dataavailability>).

12) Author contributions: You will be asked to provide CRediT (Contributor Role Taxonomy) terms in the submission system. These replace a narrative author contribution section in the manuscript.

13) A Conflict of Interest statement should be provided in the main text.

14) Every published paper now includes a 'Synopsis' to further enhance discoverability. Synopses are displayed on the journal webpage and are freely accessible to all readers. They include a short stand first (maximum of 300 characters, including space) as well as 2-5 one-sentences bullet points that summarizes the paper. Please write the bullet points to summarize the key NEW findings. They should be designed to be complementary to the abstract - i.e. not repeat the same text. We encourage inclusion of key acronyms and quantitative information (maximum of 30 words / bullet point). Please use the passive voice. Please attach

these in a separate file or send them by email, we will incorporate them accordingly.

15) Include a Reagents and Tools Table as part of the Methods section, which can be downloaded from our author guidelines (<https://www.embopress.org/page/journal/17574684/authorguide#structuredmethods>)

***** Reviewer's comments *****

Referee #1 (Comments on Novelty/Model System for Author):

This is an interesting MS that deserves to be published - I am not familiar enough with EMM to say if this journal is the best match for this study.

Referee #1 (Remarks for Author):

This work provides a comprehensive comparison between the impact of two well characterized and highly specific G4-ligands, PhenDC3 and PDS on fungal biology.

Such study is interesting, because fungal infections kill almost 4 million people annually and resistance is on the rise. Non-canonical nucleic acid structures may represent novel targets for the development of antifungals.

Quadruplexes (G4s) and i-motifs (iMs) found in the fungal pathogen *Aspergillus fumigatus* were shown to form in vitro, and formation in cells was confirmed with the QUMA-1 fluorescent probe. PhenDC3 and PDS exhibit fungistatic and fungicidal activity, respectively. PhenDC3 was as potent as some currently used antifungals and could synergise with amphotericin B. Interestingly, PhenDC3 is well tolerated by human vascular cells with negligible DNA damage. The authors identified two unusual G4s in the *cyp51A* fungal gene which were stabilised by acidic pH.

Overall, this is an interesting study that deserves to be published after taking into account the following suggestions:

- Bioinformatics: Indicate *A. fumigatus* genome size. Any information on the location of these potential G4 motifs in the genome? (promoters, etc)
- PhenDC3 is inactive against the non-Fumigati species *A. flavus* and *A. niger*. Any reason that could explain this difference? (differences in penetration? G4 content?)
- I did not initially pay attention to the fact that PhenDC3 was (sadly) inactive against biofilms (fig S1) - perhaps worth mentioning in the discussion.
- I have some minor issues with the colors and symbols chosen for figure 1 - it should be easier to distinguish the various conditions / curves. A related comment for sup Fig 6 - the yellowish color for KCl pH 5.5 does not seem to match the curves?
- Far too many significant digits in the last column of table 1 (and these values could be plotted as a bar histogram in sup info)
- Some truly weird absorbance melting spectra in sup Fig S7 - I assume this comes from normalization of a curve with little or no variation. Perhaps delete these curves or add a comment.

Referee #2 (Comments on Novelty/Model System for Author):

In general the manuscript reports the presence of G4 in fungi and the effect of G4 ligands on fungal viability. However there is no mechanistic explanation and the rationale behind the choice to investigate further two G4s is not provided. So there is a collection of data without a clear indication on the mechanisms by which the G4 ligands exert their activity. The acceptance of such data depends on the policy of the journal

Referee #2 (Remarks for Author):

In this manuscript Middleton et al present the use of two G-quadruplex (G4) ligands against fungi, including some that are resistant to antifungal compounds. They show that the two G4 ligands impact differently on the cell survival and fungistatic effect of different fungal species. They next select two genes where G4 are present and prove their folding in vitro and their interaction with the G4 ligands, suggesting possible mechanism of action and concluding that G4 ligands likely exert their activity at multiple levels, involving the many gene where G4 forming sequences have been identified.

Main comment

The reported references 12 and 13 for G4s and iMs interdependency are not correct, besides being too old.

In Materials and Methods, no indication on where the fungal strains have been obtained is present. Please provide this information.

What are the molecular bases for PhenDC3's synergy with amphotericin B, but not with itraconazole or voriconazole?

Why PhenDC3 plus amphotericin B is the only combination tested at 4 days instead of 7 days?

Supplementary Fig. 2: the text mentions that PhenDC3 and PDS were well tolerated by A549 cells. However, cells treated with PDS show sign of rounding, like they are undergoing apoptosis.

Supplementary Fig. 3: report average as in fig 5D

Table 1 reports the list of the four G4 sequences. Please add the reported location of these sequences. This is particularly important as in human cells G4 in gene promoters have been shown to exert important regulatory function on gene expression.

Have the authors investigated if a similar effect is also present in fungi?

Similarly, what was the rationale of choosing G4s in *cyp51A* and *cyp51B*? wasn't any G4 detected in promoter regions? How would G4 in coding regions affect gene expression?

Fix the following sentences:

PhenDC3 also had antifungal activity towards *A. fumigatus* comparable to current antifungals, was found to be fungistatic....

Although, colonies with a similar phenotype were recovered following treatment with the known fungicidal antifungal, amphotericin B.

Although, this is only a snapshot of some of the locations G4s may form and we estimate that there are >1000 sequences almost guaranteed to form these structures in the *A. fumigatus* genome.

Referee #3 (Remarks for Author):

In this manuscript, the authors investigate the effects of the G4-stabilizing ligands PhenDC3 and PDS on a variety of pathogenic fungi. Using simple yet definitive assays, they demonstrate that both small molecules impact the metabolic activity of a panel of fungi, though some strains proved to be more resistant than others. Overall, PhenDC3 appeared to be the more effective antifungal and had different physiological effects on the fungal cells/spores than PDS. Importantly, PhenDC3 could act synergistically with amphotericin B and had only minor effects on human cell lines at antifungal concentrations. Microscopy suggests that both PhenDC3 and PDS either stabilize or cause G4s to fold in vivo, and model physiological ssDNA sequences could fold into G4s or i-Motifs in vitro. Cumulatively, this was a straightforward and clear piece of science, and I have only a few minor issues that should be addressed before publication:

1. The G4 ligands displayed different effectiveness on the various species and strains tested. This could be due to differences in the amount of compound that enters the cell, how effectively it might be pumped out, and the number of important G4 targets it might encounter. The latter is perhaps most important if one considers the G+C richness of the various genomes. For instance, the *Saccharomyces cerevisiae* G+C content is ~38%, but G4s are nonetheless over-represented. Do the susceptible species investigated by the authors have higher G+C content or a higher predicted number of G4s in their genomes?
2. Regarding Figure 2, it is stated that PhenDC3 and PDS appear fungicidal, but at least with PhenDC3, colonies still grow after treatment. So, are the ligands actually fungicidal or fungistatic? The text is just a bit confusing here.
3. The title may be a stretch. I believe that the data presented are indicative of in vivo G4 formation but stop short of actually demonstrating in vivo G4 formation. Similarly, there is even less evidence that i-Motifs form in these cells, with only in vitro data shown using purified ssDNA. Perhaps a better title would be "Evidence that G-quadruplexes form in pathogenic fungi and represent promising antifungal targets".
4. Related to the comment above, the text states that "G4s were found to form in a range of genes involved in cell wall formation,

dynamics, and development" (pg 26). This is a misstatement. The G4s (and other structures) formed in purified ssDNA oligomers in vitro, not in the context of genes. Certainly the sequences came from physiological genomic loci, but the structure formation all happened in tubes.

**** Reviewer's comments ****

Referee #1 (Remarks for Author):

This work provides a comprehensive comparison between the impact of two well characterized and highly specific G4-ligands, PhenDC3 and PDS on fungal biology.

Such study is interesting, because fungal infections kill almost 4 million people annually and resistance is on the rise. Non-canonical nucleic acid structures may represent novel targets for the development of antifungals.

Quadruplexes (G4s) and i-motifs (iMs) found in the fungal pathogen *Aspergillus fumigatus* were shown to form in vitro, and formation in cells was confirmed with the QUMA-1 fluorescent probe. PhenDC3 and PDS exhibit fungistatic and fungicidal activity, respectively. PhenDC3 was as potent as some currently used antifungals and could synergise with amphotericin B. Interestingly, PhenDC3 is well tolerated by human vascular cells with negligible DNA damage. The authors identified two unusual G4s in the *cyp51A* fungal gene which were stabilised by acidic pH.

Overall, this is an interesting study that deserves to be published after taking into account the following suggestions:

We thank the reviewer for their positive assessment and appreciate the time taken to acknowledge the key concepts of our study.

Bioinformatics: Indicate *A. fumigatus* genome size. Any information on the location of these potential G4 motifs in the genome? (promoters, etc)

Thank you for your comment. We have previously shown that G4s can be found throughout the *A. fumigatus* genome, particularly within gene bodies and repeat regions (doi.org/10.1099/mgen.0.000570). Our understanding of promoter regions is limited in *A. fumigatus*, but G4s were not enriched in regions preceding the genes as observed in humans and *Saccharomyces* spp. This may be because the G4Hunter thresholds used to perform these recent analyses also considers G4s formed of two or more tetrads, rather than primarily focusing on those composed of three tetrads. Although two-tetrad G4s are proposed not to form intramolecular G4s, they can dimerise to form G4s and are physiologically important and often overlooked (DOI: [10.1093/nar/gkae927](https://doi.org/10.1093/nar/gkae927); <https://doi.org/10.1002/chem.202100895>). In support, it has recently been shown that G4s are enriched in the gene bodies of *Mycobacterium tuberculosis* and *Trypanosoma brucei* via G4 ChIP-seq and CUT&Tag methodologies (<https://doi.org/10.1101/2025.07.22.666098>; <https://doi.org/10.1038/s41467-025-62485-4>).

We have added a supplementary table outlining the total number of G4s per chromosome and whether they were located within the gene (including introns) or in intergenic regions. We have also added information to the manuscript to highlight this change.

Appendix Table 2. The number of predicted G4s in the *A. fumigatus* genome and whether these are in the gene bodies or intergenic regions

Chromosome	Total bp	Total G4s	G4s/kbp	G4s in gene	% G4s	Intergenic G4s	% G4s
1	4918979	7669	1.56	4898	63.87	2749	35.85
2	4844472	7870	1.62	5250	66.71	2599	33.02
3	4079167	6539	1.60	4257	65.10	2263	34.61
4	3923705	6015	1.53	3865	64.26	2138	35.54
5	3948441	6087	1.54	4069	66.85	1993	32.74
6	3778736	5811	1.54	3742	64.40	2041	35.12
7	2058334	3053	1.48	1848	60.53	1184	38.78
8	1833124	2627	1.43	1625	61.86	994	37.84

“The frequency of G4s in the *A. fumigatus* reference genome, Af293, was found to be comparable to the G4-frequency observed in humans at around 1.5 per 1000 base pairs when using the default G4Hunter settings (**Appendix Table 2**).”

We have also added the G4Hunter analysis to the methods section.

“Determination of PQS frequency and location in *Aspergillus fumigatus*”

The G4Hunter script (<https://github.com/LacroixLaurent/G4Hunter>) was used to identify putative G4s as described previously. When analysing the genomes, the scored nucleic acid sequence was computed for a sliding window of 25 nucleotides and a threshold above 1.2, which highlights sequences which are most likely to form secondary structures.”

PhenDC3 is inactive against the non-Fumigati species *A. flavus* and *A. niger*. Any reason that could explain this difference? (differences in penetration? G4 content?)

This is an interesting point and something we are currently trying to understand. We originally hypothesised that it may be due to G4-content. However, both *A. flavus* and *A. niger* have higher frequencies of putative G4-forming sequences (1.55/kbp, 1.65/kbp, and 2.46/kbp for *A. fumigatus*, *A. flavus*, and *A. niger*, respectively). Alternatively, it may be a penetration issue, as the cell wall of *A. flavus* is very different to that of *A. fumigatus* and it has been suggested to contribute to the increased drug-resistance of this species (doi: 10.3389/fcimb.2021.643312). However, our current hypothesis is that the target of PhenDC3 in *A. fumigatus* must be a G4 (or cluster of G4s) that do not exist in these other species or that the activity of drug transporters is increased/cell wall is restrictive in *A. flavus* and *A. niger*. This is something we are continuing to explore.

We have added these hypotheses to the discussion section.

“Considering we have previously shown that both *A. flavus* and *A. niger* have higher frequencies of G4-forming sequences and equivalent GC genome content to *A. fumigatus*, the loss of PhenDC3 activity is difficult to explain. However, this is suggestive that G4-ligands may be preventing the function of a specific gene (or set of genes) critical for the survival of these fungi. Indeed, through our analyses, we have found G4s to be present in

some essential genes within *A. fumigatus*, but we need to explore this further. Additionally, these species may have more active drug efflux transporters, and *A. flavus* has been shown to have a very different cell wall to *A. fumigatus* which contributes to increased resistance to cell wall-targeting drugs; meaning penetration could be an issue”

I did not initially pay attention to the fact that PhenDC3 was (sadly) inactive against biofilms (fig S1) - perhaps worth mentioning in the discussion.

Thank you for your comment. Indeed, this is unfortunate but not unexpected at the concentrations we were testing. Antifungal drugs including itraconazole, voriconazole and amphotericin B are up to 1000x less effective against *A. fumigatus* biofilms compared to planktonic spores (doi: 10.1099/jmm.0.47247-0). It is likely that activity may be observed at much higher drug concentrations, but this would come at the increased risk of off-target effects and toxicity in a clinical setting. Therefore, we chose not to explore these higher concentrations as we didn't consider them physiologically/clinically relevant. This is most likely due to a penetration issue, but it may also be due to increased drug-efflux, as this has been associated with antifungal resistance in biofilms (doi: 10.1128/AAC.01189-10). We have now discussed this in the manuscript.

“Increased drug efflux and drug penetration issues may also underlie why PhenDC3 was ineffective against mature *A. fumigatus* biofilms, as this has been observed previously for other It has also previously been observed that antifungal drugs such as the azoles and amphotericin B are 1000x less effective against *A. fumigatus* biofilms. Therefore, it may be likely that higher concentrations of PhenDC3 would be antifungal, but the increased risk of toxicity does not make this a viable therapeutic option.”

I have some minor issues with the colors and symbols chosen for figure 1 - it should be easier to distinguish the various conditions / curves. A related comment for sup Fig 6 - the yellowish color for KCl pH 5.5 does not seem to match the curves?

Thank you for raising these issues. It is always our intention to make figures as clear as possible. We have now changed how figure 1 is presented to a heatmap format and have changed the colour of the line representing KCl at pH 5.5 in Supplementary Figure 6 (now Appendix Fig. 7 since the addition of data). In addition, whilst compiling the source information for our fungal isolates, we identified that the original *A. niger* we were using was actually *A. brasiliensis*. We have repeated these experiments with our clinical *A. niger* isolate and have included this here. Activity of the compounds was comparable to that observed for *A. brasiliensis*.

Figure 1: PhenDC3 and PDS inhibit the metabolism of drug susceptible and drug-resistant fungal pathogens. Drug susceptible (22M7007854) and itraconazole-resistant (22M7004177) *A. fumigatus* clinical isolates, laboratory strain A1160+, 'wildtype' strain CEA10, and the pan azole-resistant *A. fumigatus* isolate (TR₃₄/L98H) were treated with a concentration range of (A) PhenDC3 or (B) PDS for 48 h at 37 °C. Non-*fumigatus* *Aspergillus* strains treated with (C) PhenDC3 or (D) PDS for 48 h. *Candida* species were treated with the indicated concentrations of (E) PhenDC3 and (F) PDS for 24 h. Metabolic activity was calculated as percentage of the untreated control. Experiments are in biological triplicate with each data point consisting of the mean of three technical replicates.

Appendix Fig. 7. Thermal difference spectra of the *cyp51* sequences. DNA oligonucleotides (5 μ M) were annealed in 10 mM sodium cacodylate buffer containing 100 mM LiCl, NaCl, KCl (pH 7.0), or KCl (pH 5.5) as indicated.

Far too many significant digits in the last column of table 1 (and these values could be plotted as a bar histogram in sup info)

Thank you for your comment. We have removed the decimal places and rounded to the nearest whole number. We appreciate that it looks a bit messy, but we felt that it would be beneficial to be able to see both the sequence and the fluorescence values together. We have also added the following graph as Appendix Fig. 4 to support this data.

Appendix Fig. 4. Thioflavin T fluorescence following the binding to G4s that have formed in single-stranded DNA sequences which have been annealed in a buffer containing 10 mM sodium cacodylate and 100 mM KCl. Sequences are representative of those identified in a range of genes involved in growth, virulence, and drug resistance. B-DNA unable to form a G4 was used as a negative control and the [GGGTTA]₄ telomeric repeat sequence was used as a positive control.

Some truly weird absorbance melting spectra in sup Fig S7 - I assume this comes from normalization of a curve with little or no variation. Perhaps delete these curves or add a comment.

We thank the reviewer for pointing this out. As the reviewer clearly knows, normalisation can give rise to melt curves that look strange. We wanted to present all data we analysed for transparency. We have updated the presentation of this data. Fraction folded calculations were conducted following established procedures from Mergny *et al.* (DOI: 10.1002/0471142700.nc1701s37). We have now shaded the background of the curves that did not give a transition or were not repeatable; unusual absorbance melting spectra observed in Appendixes 8 and 9 are shaded in orange. Shaded data indicate sequences that showed no repeatable thermodynamic profile, signifying the absence of clear melting or annealing transitions. We have updated the figure legend accordingly to clarify these distinctions and updated Table 2 to reflect the new changes.

Appendix Fig 8. Fraction folded UV melt and annealing profile of 5 μM of the DNA oligonucleotides A1-C, A1-G, A2-C, A2-G, B-C, and B-G at 260 nm in 10 mM sodium cacodylate buffer containing 100 mM LiCl, NaCl, KCl (all at pH 7.0) and KCl (at pH 5.5). Graphs highlighted in orange showed no repeatable thermodynamic profile at 260nm.

Appendix Fig 9. Fraction folded UV melt and annealing profile of 5 μ M of the DNA oligonucleotides A1-C, A1-G, A2-C, A2-G, B-C, and B-G at 295 nm in 10 mM sodium cacodylate buffer containing 100 mM LiCl, NaCl, KCl (all at pH 7.0) and KCl (at pH 5.5). Graphs highlighted in orange showed no repeatable thermodynamic profile at 295nm.

Referee #2 (Remarks for Author):

In this manuscript Middleton et al present the use of two G-quadruplex (G4) ligands against fungi, including some that are resistant to antifungal compounds. They show that the two G4 ligands impact differently on the cell survival and fungistatic effect of different fungal species. They next select two genes where G4 are present and prove their folding in vitro and their interaction with the G4 ligands, suggesting possible mechanism of action and concluding that G4 ligands likely exert their activity at multiple levels, involving the many gene where G4 forming sequences have been identified.

We thank the reviewer for their critical appraisal of the study. We appreciate that additional mechanistic insight would be ideal, but we feel that we have provided sufficient novel and exciting insights that are currently publishable. Teasing out these mechanistic details may take several years of work and significant monetary input that is out of scope for the current manuscript.

Main comment

The reported references 12 and 13 for G4s and iMs interdependency are not correct, besides being too old.

Thank you for noticing this. This was oversight of an issue that arose whilst updating the reference manager. This has now been amended, and we have added two different references (DOI: 10.1021/jacs.0c11708; DOI: 10.1021/jacs.6b10028).

In Materials and Methods, no indication on where the fungal strains have been obtained is present. Please provide this information.

Thank you for your comment. This information has been added to Appendix Table 1.

Appendix Table 1. The fungal isolates used in this study and minimum inhibitory concentrations

Species	Strain	MIC _{50/90} (µM)		Source
		PhenDC3	PDS	
A. fumigatus	22M7007854	0.83/1.56	6.96/12.50	This study
A. fumigatus	22M7004177	0.40/0.78	7.93/12.50	This study
A. fumigatus	CEA10	0.89/3.13	5.38/6.25	¹
A. fumigatus	A1160+	1.13/3.13	3.31/6.25	²
A. fumigatus	TR ₃₄ /L98H	0.38/0.78	5.95/12.50	³
A. hiratsukae	22M7007855	1.10/>12.50	3.98/6.25	This study
A. udagawae	22M7007856	0.69/6.25	6.09/12.50	This study
A. flavus	22M8001325	ND/ND	ND/ND	This study
A. brasiliensis	ATCC16404	ND/ND	11.43/12.50	⁴
A. niger	22M8001245	ND/ND	9.12/12.50	This study
C. albicans	JC747	ND/ND	5.94/12.5	⁵
C. auris	NCPF8985	30.27/50	5.48/6.25	⁶
C. glabrata	CBS-138	ND/ND	3.94/6.25	⁷

ND = Not determined

What are the molecular bases for PhenDC3's synergy with amphotericin B, but not with itraconazole or voriconazole?

Thank you for this insight. This is a good question and something which we are currently trying to answer. There are a couple of hypotheses surrounding this. 1) Pore formation by amphotericin B promotes the entry of PhenDC3 into cells to interact with the DNA/RNA and stabilise G4s. 2) PhenDC3 can cause downregulation of ergosterol genes similarly to the G4-binding natural product, berberine (doi: [10.3390/molecules16119218](https://doi.org/10.3390/molecules16119218)). This mechanism, shared with azoles, may explain why these two drugs do not synergise. However, if this was true, you might expect additivity or antagonism, which we did not observe. This was mentioned briefly in the discussion, but we have now added to this.

“We are not currently sure why PhenDC3 and amphotericin B synergise, but it may be that this interaction is similar to that observed for flucytosine, and the increased permeability of the cell membrane allows more PhenDC3 to enter the fungi. However, we do not observe any synergy between amphotericin B and PDS, which means that this interaction may be more complex. Further, the G4-binding natural product, berberine, has previously been shown inhibit ergosterol synthesis. This mechanism is shared with the azoles, which may explain why PhenDC3 cannot synergise with azoles, but this hypothesis is currently quite speculative”.

Why PhenDC3 plus amphotericin B is the only combination tested at 4 days instead of 7 days?

Thank you for your comment, but I think there is some confusion here, as both the figure and the text show that this combination was tested over 7-days like all the other conditions.

Supplementary Fig. 2: the text mentions that PhenDC3 and PDS were well tolerated by A549 cells. However, cells treated with PDS show sign of rounding, like they are undergoing apoptosis.

Thank you, this is a valid point, and this statement has been removed.

Supplementary Fig. 3: report average as in fig 5D

Thank you for this comment. We have attempted to quantify the number of foci for 25 μ M PDS but given the cytotoxicity of PDS in vascular smooth muscle cells (ED50 of 13.1 μ M), cells become quite permeable, and quantification becomes unreliable due to an excess of background. Similarly, 50 μ M PDS was found to frequently induce cell rupture and consequently, we could not quantify DNA damage at this concentration. We have added panel B to this Appendix figure.

Appendix Fig. 3. (A) Representative confocal microscopy images of primary human vascular smooth muscle cells treated with 25 μ M or 50 μ M PDS for 24 h. Scale bars represent 50 μ m. Cells were incubated with rhodamine phalloidin, a γ H2AX antibody, or DAPI to visualise F-actin, DNA damage, or the nucleus, respectively. Images were obtained using a Zeiss LSM 980 laser scanning confocal microscope equipped with an Airyscan 2 detector using the 20x objective. Images were analysed using Fiji 1.53t. **(B)** Quantification of the γ H2AX-foci following treatment with 25 μ M PDS.

Table 1 reports the list of the four G4 sequences. Please add the reported location of these sequences. This is particularly important as in human cells G4 in gene promoters have been shown to exert important regulatory function on gene expression. Have the authors investigated if a similar effect is also present in fungi?

Similarly, what was the rationale of choosing G4s in *cyp51A* and *cyp51B*? wasn't any G4 detected in promoter regions? How would G4 in coding regions affect gene expression?

Thank you for this important comment. The reported location of these sequences has now been added to Table 1. We have also investigated the genomic location of

these sequences throughout the entire genome, and this data has been added as supporting information (Appendix Table S2) as requested by Reviewer 1. As mentioned, promoter regions are poorly understood and defined in *A. fumigatus* but G4s are not enriched in the regions we would expect promoters to be. However, it is important to note that gene body G4s are also involved in transcriptional regulation, and this phenomenon is not only associated with G4s in promoters. Several studies have shown that G4s in gene bodies are implicated in both increased and decreased transcription. Inserting a G4 motif into the template strand in human and bacterial genes reduced gene transcription via impairment of Pol II progression, whilst a G4 on the non-template strand reduced transcription by T7 polymerase via R-loop formation (doi: 10.1021/bi401451q; doi: 10.1016/j.chembiol.2014.09.014; doi: 10.1093/nar/gkx403). Conversely, human genes with a greater number of G4s on the non-template strand had higher steady-state transcriptional levels and these are suggested to aid transcription reinitiation (doi: 10.1093/nar/gkr079).

Notably, a recent pre-print from the Di Antonio lab has shown that G4s form more readily in the gene bodies in *Trypanosoma brucei* and that PhenDC3 specifically perturbs the expression of genes bearing G4s in the coding regions (doi: 10.1101/2025.07.22.666098). Herein, we have also shown that transcription is reduced when a promoter G4 is absent in *cyp51A*.

The rationale for choosing both *cyp51A* and *cyp51B* was because they both contained sequences with the potential to form G4s and they are both significantly impacted in fungal physiology and antifungal drug resistance. Cyp51A and B are also important for survival and integrity of the cell membrane. Therefore, exploring G4s in these sequences is important to comprehensively understand fungal biology and we hypothesised that they may be involved in the phenotypes observed.

We have made amendments to the discussion to include this new evidence concerning the impact of PhenDC3 on gene body G4s.

“Unlike humans, predicted G4s were found to be enriched within the coding regions of genes in *A. fumigatus*. Recent evidence suggests that this phenomenon is not restricted to filamentous fungi and has also been observed in *Trypanosoma brucei* and *Mycobacterium tuberculosis*. The former study also demonstrated that PhenDC3 specifically perturbs the expression of genes containing G4s within their coding regions, which supports our observations with *cyp51A*. This new evidence and several previous studies indicate that gene body G4s are equally important within transcriptional control and should not be overlooked when investigating their regulatory functions.”

Fix the following sentences:

PhenDC3 also had antifungal activity towards *A. fumigatus* comparable to current antifungals, was found to be fungistatic....

Although, colonies with a similar phenotype were recovered following treatment with the known fungicidal antifungal, amphotericin B.

Although, this is only a snapshot of some of the locations G4s may form and we estimate that there are >1000 sequences almost guaranteed to form these structures in the *A. fumigatus* genome.

Thank you for your comments, but I can't find anything wrong with these sentences and I'm unsure how to fix them. I have changed a few words, hopefully this has fixed any perceived issues.

“PhenDC3 also had potent fungistatic activity, prevented germination, synergised with the antifungal amphotericin B *in vitro* and *in vivo*, and displayed low genotoxicity and cytotoxicity towards human cells.”

I have replaced 'although' with 'However' for the following two statements.

Referee #3 (Remarks for Author):

Thank you for your positive reception of our study and the time taken to provide comments.

The G4 ligands displayed different effectiveness on the various species and strains tested. This could be due to differences in the amount of compound that enters the cell, how effectively it might be pumped out, and the number of important G4 targets is might encounter. The latter is perhaps most important

if one considers the G+C richness of the various genomes. For instance, the *Saccharomyces cerevisiae* is G+C content is ~38%, but G4s are nonetheless over-represented. Do the susceptible species investigated by the authors have higher G+C content or a higher predicted number of G4s in their genomes?

This is an interesting point. Surprisingly, both species resistant to PhenDC3 have higher frequencies of putative G4-forming sequences (1.55/kbp, 1.65/kbp, and 2.46/kbp for *A. fumigatus*, *A. flavus*, and *A. niger*, respectively). Similarly, the GC content for each species is similar (49.8%, 48.3%, and 49.5% for *A. fumigatus*, *A. flavus*, and *A. niger*, respectively). Indeed, there is not a simple correlation between GC content and number of putative G4s across the entire subdivision Pezizomycotina ($R=0.2883$; doi: 10.1099/mgen.0.000570). We have also performed these analyses in a wider range of *Aspergillus* species (one species per dot in the figure) which will be submitted in a separate manuscript. However, the data can be found below for reference where we normalise the number of G4s to genome content.

We have also added an extra statement to the manuscript discussion.

“Considering we have previously shown that both *A. flavus* and *A. niger* have higher frequencies of G4-forming sequences and equivalent GC genome content to *A. fumigatus*, the loss of PhenDC3 activity is difficult to explain. However, this is suggestive that G4-ligands may be preventing the function of a specific gene (or set of genes) critical for the survival of these fungi. Indeed, through our analyses, we have found G4s to be present in some essential genes within *A. fumigatus*, but we need to explore this further. Additionally, these species may have more active drug efflux transporters, and *A. flavus* has been shown to have a very different cell wall to *A. fumigatus* which contributes to increased resistance to cell wall-targeting drugs; meaning penetration could be an issue”

Regarding Figure 2, it is stated that PhenDC3 and PDS appear fungicidal, but at least with PhenDC3, colonies still grow after treatment. So, are the ligands actually fungicidal or fungistatic? The text is just a bit confusing here.

Thank you for your comment. We found that PhenDC3 could prevent colony growth for 24 h, but after 48 h colonies arose, which was suggestive of fungistatic activity. PDS treated fungi still showed no growth at 48 h and was considered fungicidal. We have tried to clarify this in the text.

“PhenDC3 was found to be fungistatic, whilst PDS was fungicidal, with MIC values of 1.56 μM and 12.5 μM being recorded for PhenDC3 and PDS, respectively (Fig. 2A).”

The title may be a stretch. I believe that the data presented are indicative of in vivo G4 formation but stop short of actually demonstrating in vivo G4 formation. Similarly, there is even less evidence that i-Motifs form in these cells, with only in vitro data shown using purified ssDNA. Perhaps a better title would be "Evidence that G-quadruplexes form in pathogenic fungi and represent promising antifungal targets".

Thank you for the suggestion, this has been changed.

“Evidence that G-quadruplexes form in pathogenic fungi and represent promising antifungal targets”

Related to the comment above, the text states that "G4s were found to form in a range of genes involved in cell wall formation, dynamics, and development" (pg 26). This is a misstatement. The G4s (and other structures) formed in purified ssDNA oligomers in vitro, not in the context of genes. Certainly the sequences came from physiological genomic loci, but the structure formation all happened in tubes.

Thank you for this comment. As you rightly state, we measure formation in ssDNA. We have amended the text as below to reflect that we measured formation in ssDNA, rather than the genes themselves.

“G4s have the potential to form in A. fumigatus genes involved in diverse biological processes ranging from cell wall development to drug resistance

Notably, G4s were found to form in a range of purified single-stranded DNA oligonucleotides representative of sequences found in genes involved in cell wall formation, dynamics, and development; including those involved in...”

The table title has also been changed to:

“Table 1. G4s can form in sequences predicted to form G4s found in genes linked with drug resistance, virulence, and development”

14th Oct 2025

Dear Dr. Bidula,

Thank you for the submission of your revised manuscript to EMBO Molecular Medicine. I am pleased to inform you that we will be able to accept your manuscript pending the following final amendments:

1) In the main manuscript file, please do the following:

- Please address all comments suggested by our data editors listed below:

o Figure legends:

1. Please note that the exact p values are not provided in the legends of figures 3D, 6B, 8B.

2. Please note that the error bars are not defined in the legends of figures 5A, B.

- Add up to 5 keywords.

- Indicate in legends number and nature of replicates and exact p= values, not a range, along with the statistical test used. To keep the figures "clear" some authors found providing an Appendix table Sx with all exact p-values preferable. You are welcome to do this if you want to.

- In data availability statement please remove the following text "All data are available in the main text or the supplementary materials. Source data are included in this published article."

- Please correct the reference citation in the text and reference list. In the text a reference should be cited by author and year of publication. Include a space between a word and the opening parenthesis of the reference that follows. In the reference list, citations should be listed in alphabetical order. Where there are more than 10 authors on a paper, 10 will be listed, followed by "et al.". Also, please remove DOIs. Please check "Author Guidelines" for more information.

<https://www.embopress.org/page/journal/17574684/authorguide#referencesformat>

2) Appendix:

- Appendix Fig. S3 is currently too pixelated. Please replace it with a higher resolution images.

- Please correct the nomenclature to "Appendix Table S1" etc and "Appendix Figure S1" etc.

3) Synopsis:

- Please resize the image to 550 px-wide x 300-600 pixels high and upload it as a high-resolution jpeg file.

4) As part of the EMBO Publications transparent editorial process initiative (see our Editorial at <http://embomolmed.embopress.org/content/2/9/329>), EMBO Molecular Medicine will publish online a Review Process File (RPF) to accompany accepted manuscripts. This file will be published in conjunction with your paper and will include the anonymous referee reports, your point-by-point response and all pertinent correspondence relating to the manuscript. Let us know whether you agree with the publication of the RPF and as here, if you want to remove or not any figures from it prior to publication. Please note that the Authors checklist will be published at the end of the RPF.

5) Please provide a point-by-point letter INCLUDING my comments as well as the reviewer's reports and your detailed responses (as Word file).

I look forward to reading a new revised version of your manuscript as soon as possible.

Yours sincerely,

Zeljko Durdevic

Zeljko Durdevic
Senior Editor
EMBO Molecular Medicine

*** Instructions to submit your revised manuscript ***

In the event of acceptance, this file will be published in conjunction with your paper and will include the anonymous referee reports, your point-by-point response and all pertinent correspondence relating to the manuscript. If you do NOT want this file to

be published, please inform the editorial office at contact@embomolmed.org.

- 1) a .docx formatted version of the manuscript text (including Figure legends and tables)
 - 2) Separate figure files*
 - 3) supplemental information as Expanded View and/or Appendix. Please carefully check the authors guidelines for formatting Expanded view and Appendix figures and tables at <https://www.embopress.org/page/journal/17574684/authorguide#expandedview>
 - 4) a letter INCLUDING the reviewer's reports and your detailed responses to their comments (as Word file).
 - 5) The paper explained: EMBO Molecular Medicine articles are accompanied by a summary of the articles to emphasize the major findings in the paper and their medical implications for the non-specialist reader. Please provide a draft summary of your article highlighting
 - the medical issue you are addressing,
 - the results obtained and
 - their clinical impact.This may be edited to ensure that readers understand the significance and context of the research. Please refer to any of our published articles for an example.
 - 6) Author contributions: the contribution of every author must be detailed in a separate section.
 - 7) EMBO Molecular Medicine now requires a complete author checklist (<https://www.embopress.org/page/journal/17574684/authorguide>) to be submitted with all revised manuscripts. Please use the checklist as guideline for the sort of information we need WITHIN the manuscript. The checklist should only be filled with page numbers where the information can be found. This is particularly important for animal reporting, antibody dilutions (missing) and exact values and n that should be indicated instead of a range.
 - 8) Every published paper now includes a 'Synopsis' to further enhance discoverability. Synopses are displayed on the journal webpage and are freely accessible to all readers. They include a short stand first (maximum of 300 characters, including space) as well as 2-5 one sentence bullet points that summarise the paper. Please write the bullet points to summarise the key NEW findings. They should be designed to be complementary to the abstract - i.e. not repeat the same text. We encourage inclusion of key acronyms and quantitative information (maximum of 30 words / bullet point). Please use the passive voice. Please attach these in a separate file or send them by email, we will incorporate them accordingly.
- You are also welcome to suggest a striking image or visual abstract to illustrate your article. If you do please provide a jpeg file 550 px-wide x 300-600px high.
- 9) A Conflict of Interest statement should be provided in the main text
 - 10) Please note that we now mandate that all corresponding authors list an ORCID digital identifier. This takes <90 seconds to complete. We encourage all authors to supply an ORCID identifier, which will be linked to their name for unambiguous name identification.

Currently, our records indicate that the ORCID for your account is 0000-0002-3790-7138.

Please click the link below to modify this ORCID:
Link Not Available

- 11) Include a Reagents and Tools Table as part of the Methods section, which can be downloaded from our author guidelines (<https://www.embopress.org/page/journal/17574684/authorguide#structuredmethods>)

Graphs 800-1,200 DPI
Photos 400-800 DPI
Colour (only CMYK) 300-400 DPI"

*Additional important information regarding figures and illustrations can be found at <https://bit.ly/EMBOPressFigurePreparationGuideline>. See also figure legend preparation guidelines: <https://www.embopress.org/page/journal/17574684/authorguide#figureformat>

***** Reviewer's comments *****

Referee #2 (Comments on Novelty/Model System for Author):

While not immediately translatable in the clinic, the results show that G4 targeting in association with existing antifungal drugs could be an effective strategy to cure difficult-to-treat fungal infections.

Referee #2 (Remarks for Author):

The authors have adequately addressed previous comments

Referee #3 (Comments on Novelty/Model System for Author):

The model system included medically relevant fungal pathogens that were analyzed with high technical quality to produce novel results.

Referee #3 (Remarks for Author):

The authors have adequately addressed all of my concerns.

The authors addressed the remaining formatting issues.

29th Oct 2025

Dear Dr. Bidula,

We are pleased to inform you that your manuscript is accepted for publication and is now being sent to our publisher to be included in the next available issue of EMBO Molecular Medicine.

Zeljko Durdevic
Senior Editor
EMBO Molecular Medicine
